# A knowledge-guided pre-training framework for improving molecular representation learning

Han Li [1], Ruotian Zhang[1], Yaosen Min[1], Dacheng Ma[2], Dan Zhao [1] ✉ & Jianyang Zeng [1,3] ✉

Learning effective molecular feature representation to facilitate molecular property prediction is of great significance for drug discovery. Recently, there has been a surge of interest in pre-training graph neural networks (GNNs) via self-supervised learning techniques to overcome the challenge of data scarcity in molecular property prediction. However, current self-supervised learning-based methods suffer from two main obstacles: the lack of a well-defined self-supervised learning strategy and the limited capacity of GNNs. Here, we propose Knowledge-guided Pre-training of Graph Transformer (KPGT), a self-supervised learning framework to alleviate the aforementioned issues and provide generalizable and robust molecular representations. The KPGT framework integrates a graph transformer specifically designed for molecular graphs and a knowledge-guided pre-training strategy, to fully capture both structural and semantic knowledge of molecules. Through extensive computational tests on 63 datasets, KPGT exhibits superior performance in predicting molecular properties across various domains. Moreover, the practical applicability of KPGT in drug discovery has been validated by identifying potential inhibitors of two antitumor targets: hematopoietic progenitor kinase 1 (HPK1) and fibroblast growth factor receptor 1 (FGFR1). Overall, KPGT can provide a powerful and useful tool for advancing the artificial intelligence (AI)-aided drug discovery process.

The identification of molecules with desired properties presents one of the most significant challenges in the drug discovery field, given the considerable time and resources required for experimentally determining molecular properties[1,2]. In recent years, artificial intelligence (AI)-based approaches have played an increasingly pivotal role in predicting molecular properties, offering remarkable efficiency and cost-effectiveness[3–5]. One of the primary challenges of AI-based approaches for molecular property prediction is the representation of molecules[6,7]. The early machine learning-based approaches involved preliminary attempts toward representing molecules using basic handcrafted features[8–10]. Among these, the most prominent ones include molecular descriptors[11–13], which quantitatively characterize the physical and chemical profiles of small molecules, and fingerprints[14–16], which utilize binary strings to signify the presence of specific substructures within a molecular structure. The prediction methods based on these representations are highly dependent on complicated feature engineering strategies, consequently compromising their generalizability and flexibility.

Recent years have witnessed the emergence of deep learning-based methods as potentially useful tools for predicting molecular

[1]Institute for Interdisciplinary Information Sciences, Tsinghua University, 100084 Beijing, China. [2]Research Center for Biological Computation, Zhejiang Province, Zhejiang Laboratory, 311100 Hangzhou, China. [3]Present address: School of Engineering, Westlake University, Zhejiang Province, 310030 Hangzhou, China. ✉e-mail: zhaodan2018@tsinghua.edu.cn; zengjy@westlake.edu.cn

properties, primarily due to their remarkable capability of automatically extracting effective features from simple input data. Notably, a diverse range of neural network architectures, including recurrent neural networks (RNNs)[17–19], convolutional neural networks (CNNs)[20–22], and graph neural networks (GNNs)[23–26], excel in modeling molecular data across various formats, from the simplified molecular-input line-entry system (SMILES)[27] to molecular images and molecular graphs. Nevertheless, the limited availability of labeled molecules and the vastness of the chemical space have constrained their prediction performance, particularly when handling out-of-distribution data samples[6,28,29]. Along with the remarkable achievements of self-supervised learning methods in the fields of natural language processing[30,31] and computer vision[32,33], these techniques have been employed to pre-train GNNs and improve the representation learning of molecules, leading to substantial improvements in the downstream molecular property prediction tasks[28,34–42].

Current self-supervised learning methods, exemplified by GraphLoG[36], GROVER[38], and GEM[42], typically involve the modification of molecular graphs by means of node or subgraph masking, followed by the prediction of the masked components[28,38], or the utilization of contrastive learning objectives to align the modified graphs with their corresponding originals in latent space[35,37]. Molecules inherently possess characteristics tightly linked to their structures, which implies that even minor modifications of molecular graphs can lead to the loss of their semantic information. This natural characteristic of molecules potentially limits current self-supervised learning-based methods for molecular graphs to capture structural similarity among molecules and fails to capture the rich semantic information related to molecular properties encoded within their chemical structures (Supplementary Fig. 1)[43]. Moreover, in the absence of semantic information on molecular graphs, the only dependency between the masked nodes and their adjacent nodes is the valency rules, which often fail to guide the model to make accurate predictions for the masked nodes (Supplementary Fig. 2). Consequently, this limitation can potentially result in a model that simply memorizes the dataset. We hypothesize that introducing additional knowledge that quantitatively describes molecular characteristics into the self-supervised learning framework can effectively address these challenges. There are many quantitative characteristics of molecules, such as the aforementioned molecular descriptors and fingerprints, that are readily accessible through currently established computational tools[13,44]. Integrating this additional knowledge can introduce abundant semantic information about molecules into self-supervised learning, thus substantially enhancing the acquisition of semantic-enriched molecular representations.

Existing self-supervised learning methods generally rely on GNNs (e.g., graph isomorphism network[45]) as backbone models. However, GNNs can only provide limited model capacity, as they suffer from over-smoothing when increasing their numbers of layers[46,47]. Additionally, GNNs might struggle to capture long-range interactions between atoms, arising from their standard practice of exchanging information only among one-hop neighbors during the message passing process[48,49]. Recent advancements in backbone networks, particularly transformer-based models[50], have emerged as game-changers[50–52]. These models, characterized by an increasing number of parameters and the capability to capture long-range interactions, present promising avenues to comprehensively model the structural characteristics of molecules[53–57].

In this study, we introduce KPGT[58], a self-supervised learning framework designed to enhance molecular representation learning, and thus advance the downstream molecular property prediction tasks. The KPGT framework combines a high-capacity model, called Line Graph Transformer (LiGhT), particularly designed to accurately model molecular graph structures, with a knowledge-guided pre-training strategy aiming to capture both structural and semantic knowledge of molecules. After pre-training on a large-scale dataset consisting of approximately two million molecules, KPGT demonstrated a significant performance enhancement on 63 molecular property datasets. Moreover, we showcased the practical applicability of KPGT by successfully utilizing it to identify potential inhibitors for two antitumor targets, hematopoietic progenitor kinase 1 (HPK1) and fibroblast growth factor receptor (FGFR1). In summary, KPGT offers a powerful self-supervised learning framework for effective molecular representation learning, thereby advancing the field of AI-aided drug discovery.

## Results
### Overview of KPGT
Our proposed KPGT framework (Fig. 1) comprises two main components: a backbone model called Line Graph Transformer (LiGhT) and a knowledge-guided pre-training strategy. LiGhT is particularly designed to comprehensively capture the complicated patterns within molecular graph structures (Fig. 1b). This model builds upon a classic transformer encoder[50], consisting of multiple layers of a multi-head attention module and a feed-forward network. LiGhT takes the molecular line graphs as input, which represent the adjacencies between edges of the original molecular graphs (Supplementary Fig. 3). Representing molecules as line graphs allows LiGhT to fully take advantage of the intrinsic features of chemical bonds, which are generally neglected in the previously defined transformer architectures[53–57]. Moreover, in order to precisely model the structural information of molecules, we introduce two positional encoding modules, namely distance encoding and path encoding modules, into the multi-head attention module.

Our proposed knowledge-guided pre-training strategy is based on a masked graph model objective[59], which initially masks a subset of nodes in molecular graphs at random and subsequently learns to predict these masked nodes (Fig. 1a). The most distinguishing feature of our strategy is the incorporation of additional knowledge. Each molecular graph is augmented with a knowledge node (K node) connected to the original nodes within the graph. The raw feature embedding of each K node is initialized using the corresponding additional knowledge. During pre-training, the K node interacts with other nodes in the multi-head attention module of each transformer layer, thereby providing guidance for predicting the masked nodes. This mechanism enables the backbone model to effectively capture both structural and semantic information within molecular graphs.

We utilize around two million molecules from the ChEMBL29 dataset[60] to pre-train LiGhT using the knowledge-guided pre-training strategy. We then apply transfer learning to the pre-trained LiGhT model to carry out downstream molecular property prediction tasks. A multiple-layer perceptron is integrated on top of the LiGhT model to serve as a predictor. The transfer learning approach can be classified into two settings based on whether the parameters of the pre-trained LiGhT model are trainable: finetuning (Fig. 1c) and feature extraction (Fig. 1d). In the finetuning setting, we introduce several finetuning strategies, such as layer-wise learning rate decay[61], re-initialization[61], FLAG[62], and $L^2$-SP[63], thereby fully taking advantage of the knowledge acquired by the pre-trained model. Further details about the KPGT framework can be found in the "Methods" section.

### KPGT outperforms the baseline methods on molecular property prediction
We initially compared KPGT to 19 state-of-the-art self-supervised learning-based methods (Supplementary Note 1.2) on 11 molecular property datasets. Among these datasets, eight are designed for classification tasks, while the remaining three are designed for regression tasks, collectively spanning a wide range of molecular properties, such as biophysics, physiology, and physical chemistry. The comparison was conducted under two settings: feature extraction, where the backbone model was fixed, and finetuning, where the backbone model

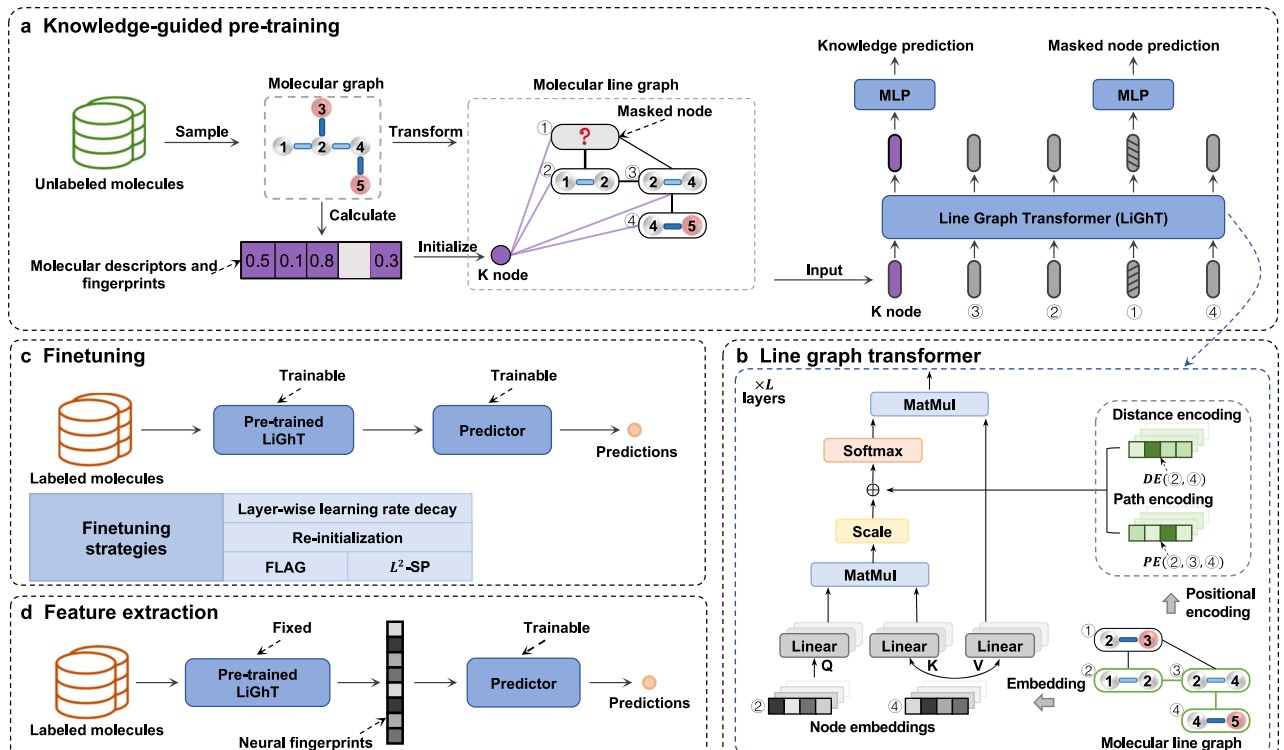

**Fig. 1 | An illustrative diagram of KPGT. a** A knowledge-guided pre-training strategy based on a masked graph model and enhanced by additional knowledge. Molecules are represented as molecular line graphs, which represent the adjacencies between the edges of the original molecular graphs. **b** A line graph transformer based on classic transformer architecture. **c** Transfer learning for downstream molecular property prediction under the finetuning setting, where the parameters of the pre-trained LiGhT are trainable. Various finetuning strategies, such as layer-wise learning rate decay, re-initialization, FLAG, and $L^2$-SP are introduced in this setting. **d** Transfer learning for downstream molecular property prediction under the feature extraction setting, where the parameters of the pre-trained LiGhT are fixed. The neural fingerprints represent the molecular feature representations generated by the pre-trained LiGhT, serving as informative and discriminative representations of molecules. Multi-layer perceptron (MLP), linear layer (Linear), matrix multiplication (MatMul), distance encoding (DE) module, path encoding (PE) module.

was trainable. Detailed experimental settings are available in the "Methods" section. In the feature extraction setting, KPGT exhibited superior performance compared to baseline methods across seven out of eight classification datasets and two out of three regression datasets, leading to overall relative improvements of 2.0% for classification and 4.5% for regression (Fig. 2a and Supplementary Tables 4 and 5). In the finetuning setting, KPGT outperformed baseline methods across seven out of eight classification datasets and all three regression datasets, resulting in overall relative improvements of 1.6% for classification and 4.2% for regression (Fig. 2b and Supplementary Tables 6 and 7). These results demonstrated that KPGT presents a more powerful self-supervised learning framework in comparison with previous methods for molecular representation learning. Moreover, among these baseline methods, GROVER[38], Supervised+Context Prediction (Contextpred$_{Sup}$)[28], Supervised+Attribute Masking (Masking$_{Sup}$)[28], Supervised+Edgepred (Edgepred$_{Sup}$)[28], and Supervised+Infomax (Infomax$_{Sup}$)[28] also integrate additional knowledge by introducing supervised graph-level pre-training tasks in their self-supervised learning frameworks, such as predicting the presence of molecular motifs or bio-activities of molecules. Although the self-supervised learning strategies of these methods are not well-defined, they still outperformed other baseline methods that do not incorporate additional knowledge. This observation demonstrated the significance of integrating additional knowledge to enhance the efficacy of pre-training on molecules.

Next, we compared the performance of KPGT with machine learning and supervised deep learning-based methods using the datasets from Therapeutics Data Commons (TDC)[64,65] and

MoleculeACE[66]. TDC is a benchmarking platform that encompasses a comprehensive set of 22 molecular property prediction tasks[64,65]. These tasks span a broad spectrum of molecular properties, including absorption, distribution, metabolism, excretion, and toxicity (ADMET), which are pivotal in the field of drug discovery and development. We conducted a comparison between KPGT and 28 baseline methods from the TDC leaderboards. These baselines included 16 deep learning-based methods and 12 machine learning-based methods (Supplementary Note 1.2). Detailed experimental settings are available in the "Methods" section. Based on the results, KPGT exhibited superior performance compared to the baseline methods on 16 out of 22 datasets. Specifically, it outperformed the baselines on five out of six absorption datasets, two out of three distribution datasets, six out of six metabolism datasets, one out of three excretion datasets, and two out of four toxicity datasets (Fig. 2c). These findings showcased the capability of KPGT to provide robust and generalizable molecular representations, making it versatile in predicting diverse aspects of molecular properties. More comprehensive results are available in Supplementary Figs. 4−8 and Supplementary Table 8.

We also assessed the performance of KPGT on more challenging tasks: predicting bio-activities for activity cliffs. Activity cliffs refer to pairs of molecules exhibiting highly similar structures but displaying substantial difference in potency, posing a significant challenge for prediction models. MoleculeACE is a benchmarking platform that provides 30 bio-activity datasets involving activity cliffs, derived from 30 macromolecular targets[66]. In this evaluation, we assessed KPGT against 24 baseline methods from the MoleculeACE benchmarking platform, comprising seven deep learning-based methods and 17

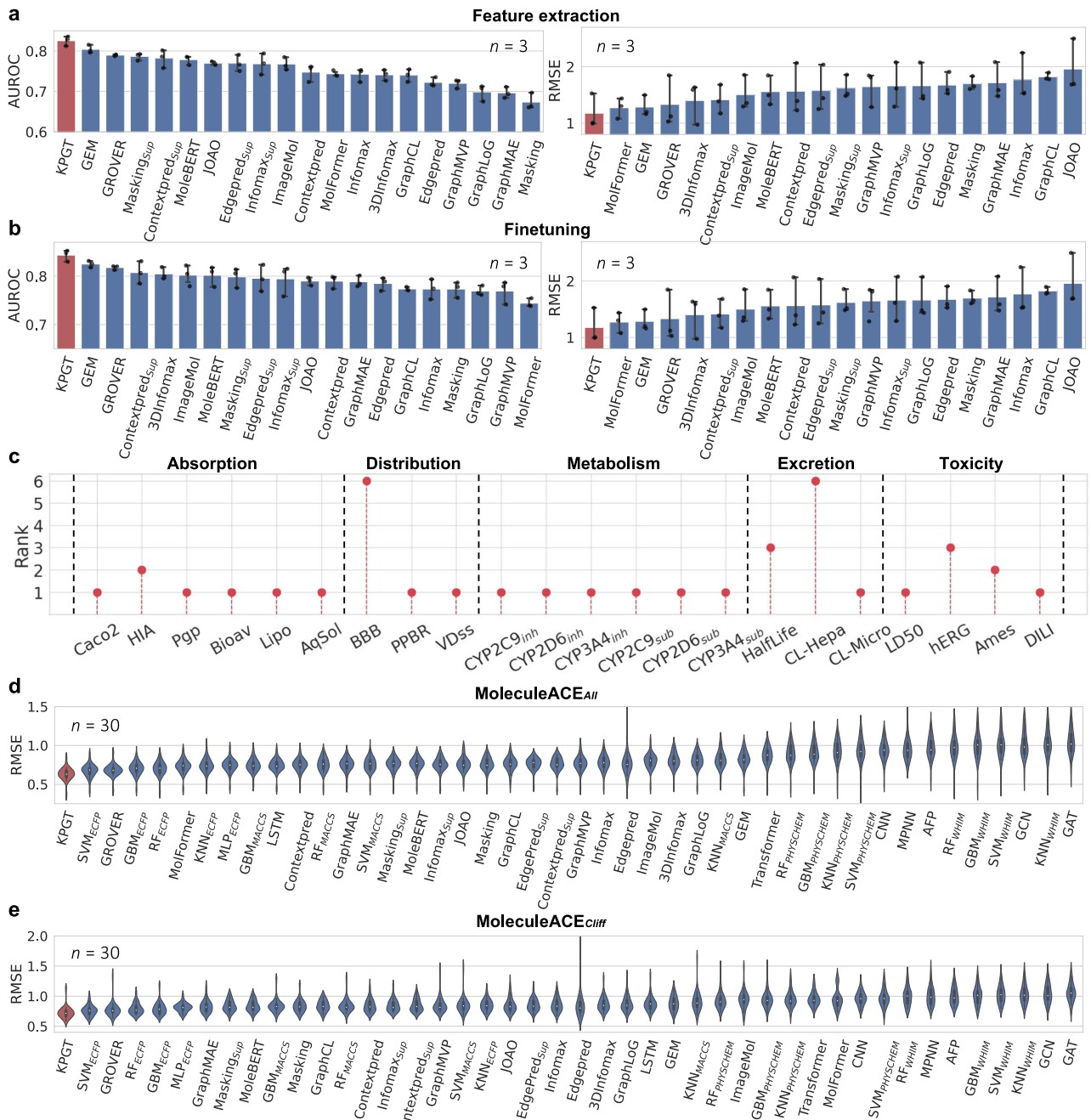

**Fig. 2 | Comparison assessments of KPGT and baseline methods on molecular property prediction. a, b** The averaged results of KPGT and self-supervised learning-based baseline methods on eight classification datasets (measured in terms of AUROC) and three regression datasets (measured in terms of RMSE), under the feature extraction setting (where the backbone model was fixed) and the finetuning setting (where the backbone model was trainable). The results were reported based on three independent runs with different random seeds. Data are presented as mean ± standard deviation (SD). **c** The ranking results of KPGT on the leaderboards from the TDC benchmarking platform for predicting absorption, distribution, metabolism, excretion, and toxicity properties of molecules. The ranking results were reported based on the averaged results derived from five independent runs. Dashed lines are used to separate different molecular property categories. **d, e** The averaged results of KPGT and baseline methods on the 30 bio-activity datasets from MoleculeACE, measured in terms of RMSE on all molecules from the test sets (denoted as MoleculeACE$_{All}$) and solely on activity cliffs from the test sets (denoted as MoleculeACE$_{Cliff}$), respectively. The violin plot displays the minima and maxima (lower and upper ends), median (the white dot in the center), interquartile range (lower and upper ends of the box), and 1.5 times of interquartile range (whiskers). Source data are provided as a Source Data file.

machine learning-based methods (Supplementary Note 1.2), as well as all 19 self-supervised learning-based methods from our first benchmarking test. Detailed experimental settings are available in the "Methods" section. Fig. 2d and Supplementary Table 9 report the results of KPGT and baseline methods. The results demonstrated that

KPGT outperformed baseline methods on 26 out of 30 datasets, with an overall relative improvement of 3.9%. Figure 2e and Supplementary Table 10 present the results of KPGT and baseline methods evaluated exclusively on the activity cliffs within each test set. From the results, KPGT achieved superior prediction performance in comparison with

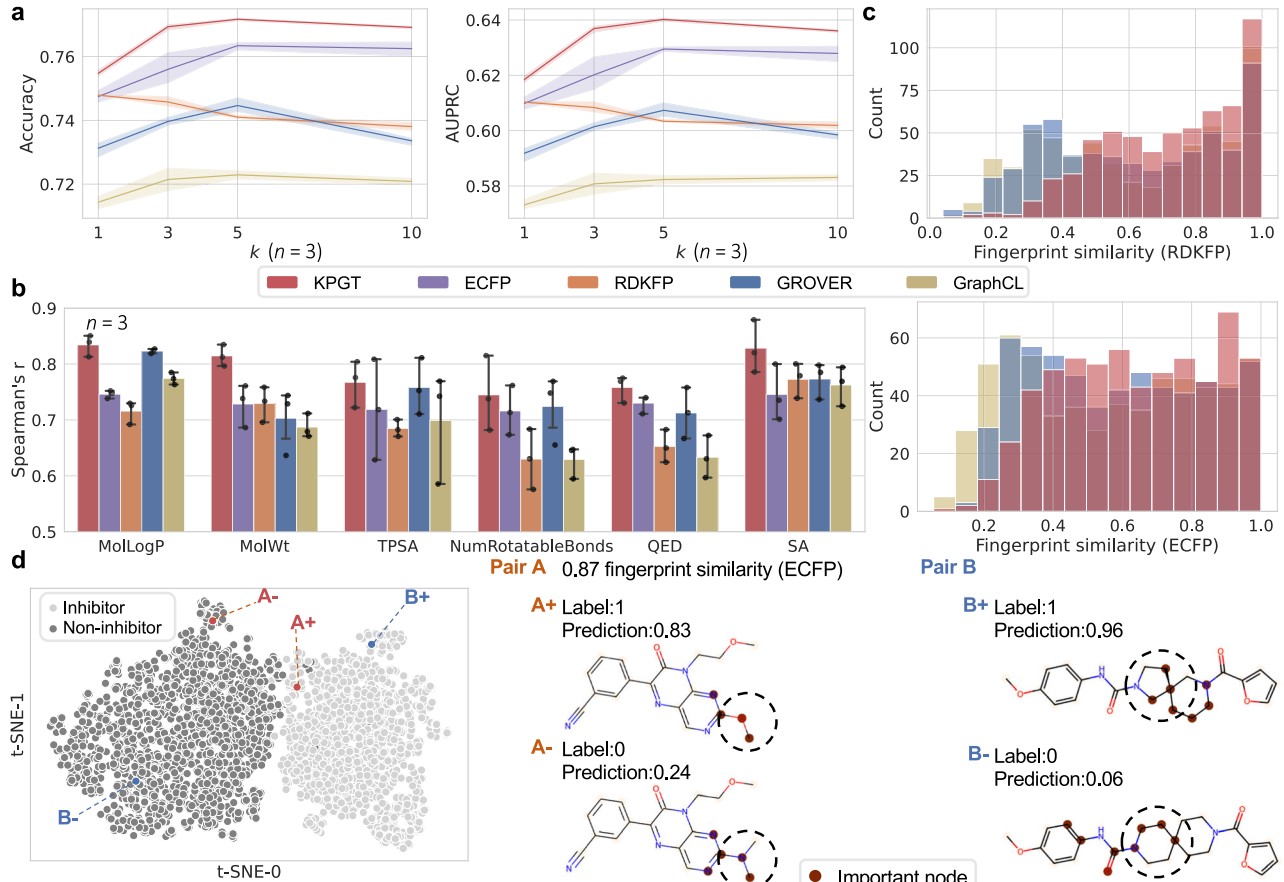

**Fig. 3 | Investigating the knowledge acquired by KPGT in pre-training and finetuning. a** The accuracy and AUPRC of $k$-nearest neighbor classification for predicting drug metabolism on the CYP3A4 dataset given different values of $k$. The error bands stand for the standard deviations across three independent runs. **b** The Pearson's r between five descriptors (i.e., MolLogP, MolWt, TPSA, NumRotatableBonds, QED, and SA) of 200 target molecules and their corresponding closest molecules identified using fingerprints from KPGT and the baseline methods. The results were based on three independent runs. Data are presented as mean ±

standard deviation (SD). **c** The distribution of the structural similarity of 200 target molecules and their corresponding closest molecules, which were identified using the fingerprints from KPGT and the baseline methods. **d** t-SNE visualization of molecular representations from the CYP3A4 dataset produced by KPGT, accompanied by the visualization of critical nodes in activity cliffs identified by SubgraphX[71]. Dashed circles highlight the distinguished substructures within the activity cliffs. Source data are provided as a Source Data file.

baseline methods on 22 out of 30 datasets, exhibiting an overall relative improvement of 1.2%. These observations demonstrated the efficacy of KPGT in predicting molecular bio-activities, even when handling activity cliffs.

Additionally, the results obtained on the datasets from TDC and MoleculeACE revealed that traditional machine learning methods based on handcrafted features (e.g., molecular descriptors and fingerprints) exhibited highly competitive prediction performance, even surpassing the previously designed deep learning-based methods in multiple cases. For the datasets from TDC, machine learning-based methods served as the most competitive baseline methods on 13 out of 22 datasets (Supplementary Table 8). Similarly, for the datasets from MoleculeACE, machine learning-based methods emerged as the most competitive baseline methods on 18 out of 30 datasets (Supplementary Tables 9 and 10). These findings suggested that the previous deep learning-based models may struggle to capture the intrinsic features of molecules, which aligned with observations of earlier studies[67,68]. Significantly, our experimental findings consistently showcased the superior performance of KPGT, surpassing these machine learning-based methods by a substantial margin. In summary, the aforementioned discoveries provided compelling evidence that KPGT consistently outperformed baseline methods on a total of 63 datasets, thereby demonstrating its efficacy and reliability in the prediction of molecular properties.

## Investigating the knowledge acquired by KPGT in pre-training and finetuning

After demonstrating the superiority of KPGT in predicting molecular properties, we conducted further investigations to unveil the reasons underlying its remarkable performance. We initiated this exploration by analyzing the latent space constructed by KPGT after pre-training. For this analysis, we employed a dataset comprising measurements of bio-activity (i.e., inhibition or non-inhibition) for 12,328 molecules targeting the CYP3A4 enzyme, which plays a crucial role in drug metabolism[69]. We first generated neural fingerprints (i.e., molecular feature representations) for molecules from the CYP3A4 dataset using the pre-trained KPGT. Next, we randomly sampled 200 molecules from the CYP3A4 dataset to form a test set, while the remaining molecules constituted the training set. We then employed k-nearest neighbor classification (kNN) based on the neural fingerprints to make predictions for the molecules in the test set. We compared the performance of KPGT with two classic fingerprints widely used in the field, namely extended-connectivity fingerprints (ECFP)[14] and RDKit fingerprint (RDKFP)[44], along with two neural fingerprints derived from two state-of-the-art self-supervised learning-based methods, GROVER[38] and GraphCL[35]. Figure 3a illustrates the comparison performance of KPGT and baseline methods in terms of accuracy and the area under Precision-Recall (AUPRC). Notably, KPGT achieved relative improvements of 1.3%-2.7% over baseline methods in terms of AUPRC. This

result indicated that neighboring molecules in the latent space constructed by the pre-trained KPGT may tend to establish more similar characteristics.

We carried out additional tests to further verify our findings. More specifically, we first retrieved the closest data point in the latent space for each molecule from the sampled test set, yielding 200 pairs of molecules. Next, we calculated five descriptors that play important roles in drug discovery, including molecular LogP (MolLogP), molecular weight (MolWt), topological polar surface area (TPSA), number of rotatable bonds (NumRotatableBonds), quantitative estimate of drug-likeness (QED), and synthetic accessibility (SA), for the sampled molecules and queried ones. We then measured the correlation between the pairs of sampled and queried molecules for each descriptor in terms of Spearman's rank correlation coefficient (Spearman's r) and Pearson correlation coefficient (Pearson's r). Figure 3b and Supplementary Fig. 9 report the results of KPGT and baseline methods, which indicated that KPGT achieved higher correlations on these five descriptors compared to baseline methods. Additionally, we measured the structural similarities between the sampled molecules and the corresponding queried ones by calculating the Tanimoto similarity between their RDKFP/ECFP fingerprints. From the results, the molecules queried by KPGT exhibited higher structural similarity in comparison with the self-supervised learning-based baseline methods (Fig. 3c and Supplementary Fig. 10). These results collectively demonstrated that in the latent space learned by KPGT, the proximity of molecules not only implied similarity in their structures but also signified similar semantics. This finding suggested that our proposed knowledge-guided pre-training strategy enabled LiGhT to effectively capture both structural and semantic information of molecules, thus providing sufficient knowledge for the downstream molecular property prediction tasks.

Next, we proceeded with the finetuning process of KPGT utilizing the CYP3A4 dataset. To elaborate, we finetuned KPGT on the CYP3A4 dataset and employed t-distributed stochastic neighbor embedding (t-SNE)[70] to visualize the molecular representations within the test set. From the results, KPGT provided a clear separation for the representations of inhibitors and non-inhibitors, indicating that KPGT can learn the distinguishable features of molecules with different properties in the finetuning process (Fig. 3d and Supplementary Fig. 11). Subsequently, we assessed the ability of KPGT to classify activity cliffs within the CYP3A4 dataset. Notably, KPGT outperformed the baseline methods with a relative improvement of 7.1% in terms of AUPRC (Supplementary Fig. 12). To gain deeper insights into the predictions made by KPGT, we visualized two pairs of activity cliffs in the latent space, denoted as pairs A+ and A−, and pairs B+ and B−, respectively. As illustrated in Fig. 3d, despite possessing high fingerprint similarity (0.87 and 0.96), the activity cliffs were correctly located in the latent space. These results highlighted the remarkable sensitivity of KPGT in capturing the semantic changes of molecules during the finetuning process, even in cases where such changes arose from subtle structural alterations. For a more comprehensive interpretation of the predictions, we employed a graph-based explanation method named SubgraphX[71] to visualize the activity cliffs and provided interpretation for the predictions (Fig. 3d). Our observations revealed that KPGT successfully captured the key substructures that distinguished the activity cliffs, demonstrating its ability to identify the discriminative features of molecules with different properties in the finetuning stage and thus provide meaningful interpretability for its predictions.

### Uncovering potentially effective inhibitors for antitumor targets by KPGT

Hematopoietic progenitor kinase 1 (HPK1) and fibroblast growth factor receptor (FGFR1), which are implicated in a variety of cancer types, have been extensively studied for antitumor therapy[72–75]. The availability of high-quality experimental data for HPK1 and FGFR1 significantly facilitates the development and validation of the AI-based computation model, providing adequate data for evaluating the practicality and prediction performance of KPGT. In this section, we carried out evaluation tests, drug repurposing, and docking analyses for both targets, serving as a proof-of-concept validation of the effectiveness of KPGT in real-world drug discovery scenarios.

To facilitate the identification of potent inhibitors against HPK1, we collected 4442 molecules with experimentally determined potency against HPK1 from previous patents and research, measured in terms of the negative logarithm of the half maximal inhibitory concentration (pIC50). We comprehensively evaluated the prediction performance of KPGT on this dataset using three distinct splitting approaches, including scaffold splitting, time splitting, and domain transfer ("Methods"). Comparison results between KPGT and 19 self-supervised learning-based baseline methods are detailed in Fig. 4b–d and Supplementary Fig. 14. The results demonstrated that KPGT significantly outperformed 19 self-supervised learning-based baseline methods in terms of Spearman's r and Pearson's r. Remarkably, even in the time splitting and domain transfer scenarios, where the molecules within training and test sets were significantly different in their structures (Fig. 4a), KPGT consistently achieved elevated correlation scores. These observations validated the superior generalizability and reliability of KPGT in the prediction of HPK1 inhibitors.

We next sought to use KPGT to identify potential HPK1 inhibitors through drug repositioning. More specifically, we first obtained 2718 US Food and Drug Administration-approved (FDA) drugs (denoted as the FDA dataset) collected from DrugBank[76]. Then we finetuned KPGT on the pIC50 dataset of HPK1 inhibitors and made predictions for the molecules from the FDA dataset. Supplementary Table 11 reports the experimental evidence from previous studies for the top 20 predictions of KPGT. The results revealed that 12 out of 20 drugs were experimentally validated by previous assays as potential inhibitors of HPK1. For example, sunitinib, identified by KPGT, is a multi-receptor tyrosine kinase (RTK) inhibitor and a previous study reported its Ki value against HPK1 kinase at approximately 16 nM through competition binding assays[77].

To strengthen our findings, we further conducted docking analyses for the top 20 predictions from KPGT. Autodock Vina[78,79], a widely used docking software, was employed for these tests. The reference protein-ligand structure (PDB ID: 7SIU[80]) guided the identification of the binding pocket. As depicted in Fig. 4f, all the molecules achieved docking energies below −7 kcal/mol, a commonly used threshold for drug-like molecules[81–84], signifying the substantial potential for these molecules as HPK1 inhibitors. Additionally, we conducted an in-depth analysis of the protein-ligand interactions for all the molecules that had not been reported in the literature using a widely applied protein-ligand interaction profiler named PLIP[85]. Figure 4g illustrates the protein-ligand interaction profile of the ligand gilteritnib with the protein HPK1. The analysis revealed the formation of three hydrophobic interactions and six hydrogen bonds between the ligand and the protein. Remarkably, the hydrogen bonds formed with residues 94A and 97A were also reported in the reference protein-ligand structure (PDB ID: 7SIU[80]). These observations showcased that the molecules can tightly bind to HPK1, validating the reliability of the docking results. Supplementary Fig. 16 provides additional protein-ligand interaction profiles for other molecules, including palbociclib, ripretinib, trilaciclib, rucaparib, selpercatinib, alatrofloxacin, and vericiguat. Collectively, these results highlighted the superior ability of KPGT to identify potential inhibitors for HPK1.

Next, as a second prototypical example, we conducted tests on FGFR1, another promising druggable target associated with tumor progression and invasion[74,75]. We first collected 12,461 existing molecules for FGFR1 from patents and previous studies with experimental pIC50 values. We evaluated the prediction performance of KPGT on the FGFR1 dataset under the scaffold splitting and time splitting

 

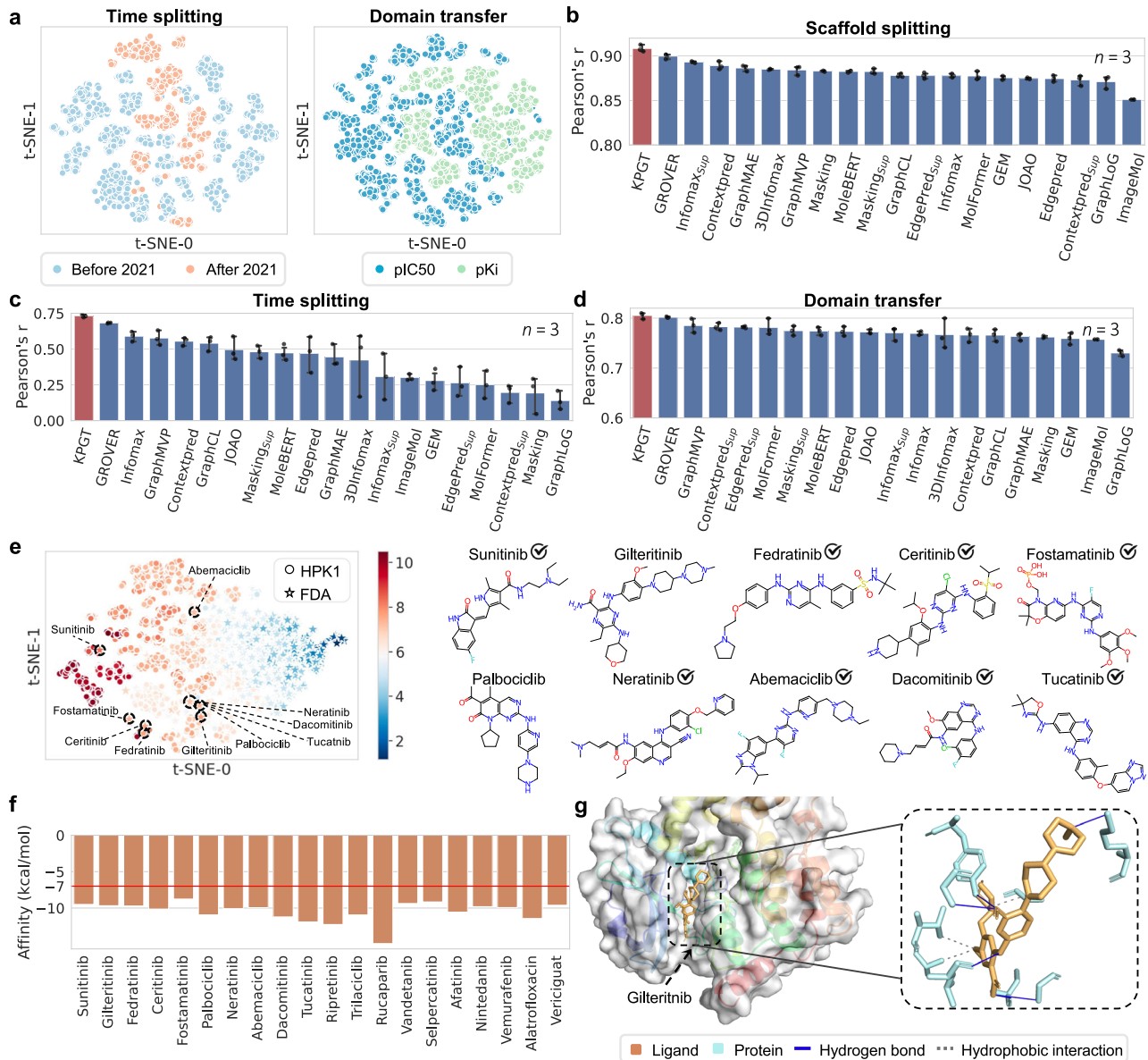

**Fig. 4 | Identifying HPK1 inhibitors using KPGT. a** Visualization of molecular ECFPs via t-SNE in the time splitting and domain transfer scenarios, respectively. **b**–**d** The performance of KPGT and baseline methods on predicting HPK1 inhibitors in the **b** scaffold splitting, **c** time splitting, and **d** domain transfer scenarios, measured in terms of Pearson's r. All the prediction results were reported based on three independent runs. Data are presented as mean ± standard deviation (SD). **e** Visualization of molecular representations of molecules from the HPK1 pIC50 dataset and the FDA dataset derived from KPGT. The top ten predictions of potential inhibitors against HPK1 from the FDA dataset derived from KPGT are delineated in dashed circles, and their corresponding molecular structures are listed in the right panel. The check symbols indicate that the molecules had been previously identified as inhibitors of HPK1 in previous studies. **f** The docking scores of the top 20 molecules identified by KPGT as potential inhibitors against HPK1, measured by Autodock Vina[78,79]. **g** The interactions between gilteritnib and HPK1 profiled by PLIP[85]. The protein-ligand structure (PDB ID: 7SIU[80]) was utilized as a reference for binding pocket identification. The red line at −7 kcal/mol represents a commonly used threshold for identifying drug-like molecules. Source data are provided as a Source Data file.

settings. Figure 5a, b and Supplementary Fig. 15 illustrate the performance of KPGT and 19 self-supervised learning-based baseline methods. KPGT achieved high correlation values under both scaffold splitting (Pearson's r = 0.924) and time splitting (Pearson's r = 0.716) scenarios. Next, we carried out drug repurposing utilizing the FDA dataset. The results revealed that 13 out of 20 top predicted small molecules were experimentally validated by previous studies as high affinity or effective FGFR1 inhibitors (Supplementary Table 12). For the docking tests, the protein-ligand structure (PDB ID: 5A4C[86]) was employed as a reference for the binding pocket identification. As shown in Fig. 5d, all the top 20 molecules identified by KPGT achieved docking energies below -7 kcal/mol. By profiling the protein-ligand interactions utilizing PLIP[85], the ligand brigatinib was tightly bound to the protein FGR1. Specifically, it formed four hydrophobic interactions, one hydrogen bond, and one salt bridge with FGFR1 (Fig. 5e). Among these interactions, the hydrogen bond formed with residue 641A was also reported in the reference protein-ligand structure (PDB ID: 5A4C[86]). In Supplementary Fig. 17, we also displayed additional protein-ligand interaction profiles for other molecules, including ripretinib, encorafenib, elagolix, baricitinib, enasidenib, and ruxolitnib. All these observations collectively reinforced the generalizability of KPGT in accelerating the identification of potential drug candidates, thus establishing its utility as a valuable tool in drug discovery.

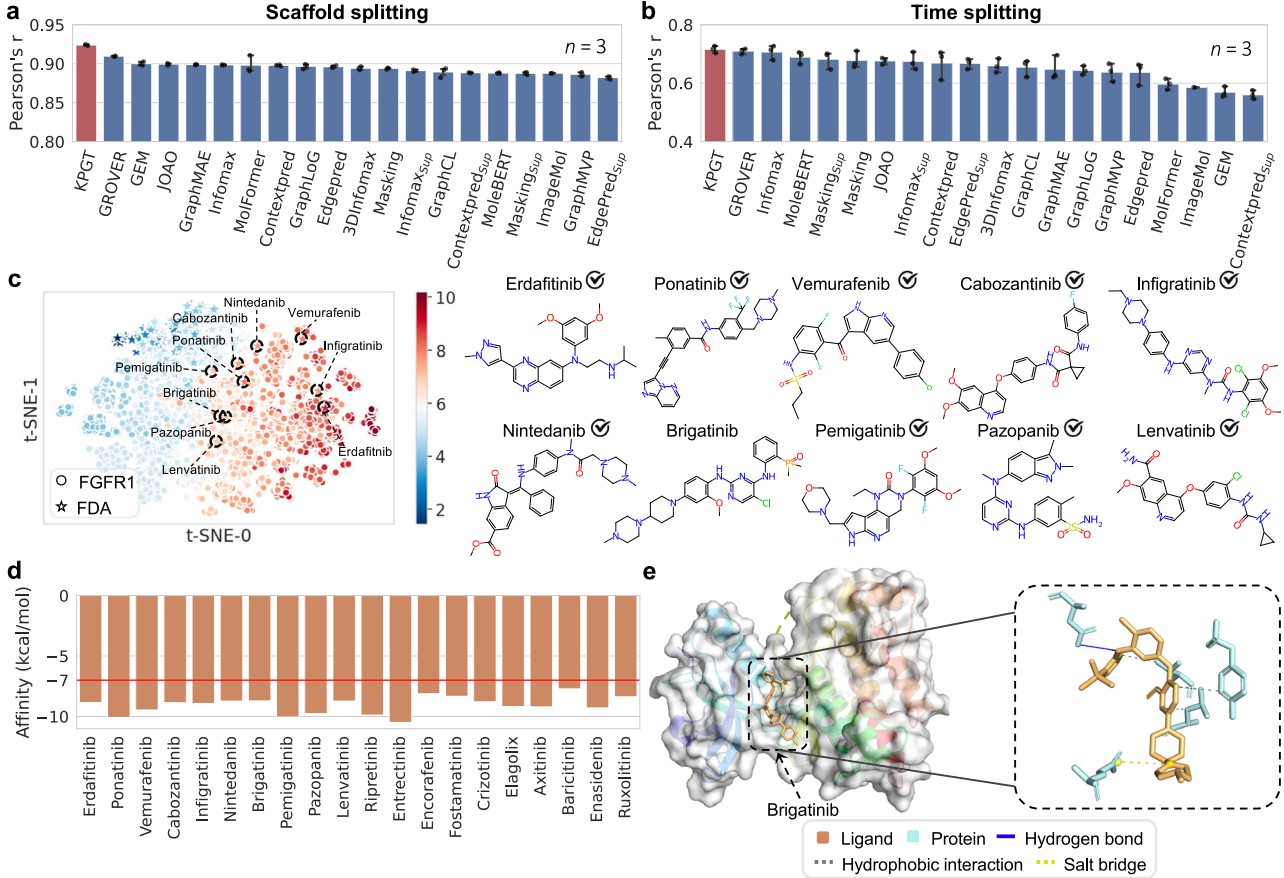

**Fig. 5 | Identifying FGFR1 inhibitors using KPGT. a, b** The performance of KPGT and baseline methods on predicting FGFR1 inhibitors under the **a** scaffold splitting and **b** time splitting scenarios, respectively, measured in terms of Pearson's r. All the prediction results were reported based on three independent runs. Data are presented as mean ± standard deviation (SD). **c** Visualization of molecular representations of molecules from the pIC50 dataset of FGFR1 inhibitors and FDA dataset derived from KPGT. The top ten predictions of potential inhibitors against FGFR1 from the FDA dataset derived from KPGT were delineated in dashed circles, and their corresponding structures were listed in the right panel. The check symbols indicate that the molecules had been previously identified as inhibitors of FGFR1 in previous studies. **d** The docking scores of the top 20 molecules identified by KPGT as potential inhibitors against FGFR1, measured by Autodock Vina[78,79]. **e** The interactions between brigatinib and FGFR1 profiled by PLIP[85]. The protein-ligand structure (PDB ID: 5A4C[86]) was utilized as a reference for binding pocket identification. The red line at −7 kcal/mol represents a commonly used threshold for identifying drug-like molecules. Source data are provided as a Source Data file.

## Ablation studies on KPGT

To validate the effectiveness of the specific design choices of KPGT, we conducted comprehensive ablation studies. Specifically, we introduced several modified frameworks based on KPGT with specific restrictions: KPGT-Pretrain (without pre-training), KPGT-KN (without knowledge nodes), KPGT-PE (without path encoding module), KPGT-DE (without distance encoding module), KPGT-LG (replacing molecular line graph with original molecular graph), KPGT-LiGhT+Graphormer (replacing backbone model LiGhT with Graphormer[57]), and KPGT-LiGhT+GIN (replacing backbone model LiGhT with GIN[45]). Additionally, for a fair comparison, we ensured that all models had approximately the same number of parameters (around 3.5 million). Supplementary Table 13 reports the performance of KPGT and modified frameworks on the datasets from our first benchmarking test. The results showcased the superiority of KPGT over KPGT-LG, resulting in an overall relative improvement of 2.3%. This observation demonstrated the enhanced informativeness of the molecular line graph utilized in KPGT, in contrast to the original molecular graph used in previous studies. KPGT also outperformed KPGT-LiGhT+Graphormer with an overall relative improvement of 1.8%, indicating the enhanced capacity of our proposed backbone model, LiGhT, in effectively capturing the inherent structural information in comparison with Graphormer. Moreover, KPGT surpassed KPGT-PE and KPGT-DE with

overall relative improvements of 2.7% and 3.0%, respectively, providing empirical validation for the significant role played by the distance encoding and path encoding modules. Overall, KPGT outperformed all the modified frameworks, highlighting the significant contributions of the individual design components of KPGT to its superior performance.

We also investigated the effect of different masking rates on the prediction performance of KPGT. We individually pre-trained KPGT with masking rates of 15, 30, 50, and 60%, and subsequently evaluated their prediction performance. From the results, setting the masking rate to 50% achieved the best prediction performance (Supplementary Fig. 19)[58]. This optimal masking rate was notably higher than that utilized in the previous methods (15%). This discrepancy indicated that the additional knowledge incorporated in KPGT effectively guided the model to predict the masked nodes, thereby enabling it to capture the rich semantics of molecules. Moreover, this observation aligned with the finding in the field of CV that applying larger masking rates in self-supervised learning can yield improved performance on downstream tasks[32].

## Discussion

In this study, we develop and establish KPGT, a self-supervised learning framework that offers improved, generalizable, and robust molecular

property prediction through significantly enhanced molecular representation learning. By leveraging a high-capacity backbone model called LiGhT, KPGT comprehensively captures the inherent structural information within molecular graphs. More importantly, KPGT introduces a knowledge-guided pre-training strategy that can robustly address the limitations of previously ill-defined pre-training approaches, empowering our model to provide semantic-enriched molecular representations. In addition, KPGT incorporates several finetuning strategies that effectively integrate the acquired knowledge from the pre-trained model, leading to improved performance on downstream molecular property prediction tasks. With these advancements, KPGT achieved great improvement on 63 datasets compared with several baseline methods. Remarkably, KPGT showed practical applications in the identification of multiple potential inhibitors for two antitumor targets.

Despite the advantages of KPGT for effective molecular property prediction, there remain a few limitations. First, the integration of additional knowledge is the most distinguishable feature of our proposed method. Except for the 200 molecular descriptors and 512 RDKFP employed in KPGT, there is potential to incorporate various other types of additional informative knowledge. For instance, Mordred[13] can calculate over 1800 molecular descriptors, presenting an opportunity to incorporate a broader range of knowledge to further enrich the representation learning of molecules. Moreover, further studies could encompass the integration of three-dimensional (3D) molecular conformations into the pre-training process, thus enabling the model to capture vital 3D information regarding molecules, and potentially enhancing the representation learning capabilities. Additionally, while KPGT currently employs a backbone model with approximately one hundred million parameters, along with pre-training on two million molecules, exploring larger-scale pre-training could offer even more substantial benefits for molecular representation learning. Overall, we anticipate our proposed method will offer a general self-supervised learning framework for accelerating AI-aided drug discovery.

## Methods

### Molecular graph construction

Given a SMILES representation of a molecule, we first abstract it as a molecular graph $\mathcal{G} = (\mathcal{V}, \mathcal{E})$, where $\mathcal{V} = \{v_i\}_{i \in [1, N_v]}$ stands for the set of nodes (i.e., atoms), $\mathcal{E} = \{e_{i,j}\}_{i,j \in [1, N_v]}$ stands for the set of edges (i.e., chemical bonds), and $N_v$ stands for the number of nodes. We initialize the features of nodes and edges in the molecular graph via RDKit[44]. The atom and bond features used in this work are summarized in Supplementary Tables 1 and 2, respectively. We represent the initial features of node $v_i$ and edge $e_{i,j}$ as $\boldsymbol{x}_i^v \in \mathbb{R}^{D_v}$ and $\boldsymbol{x}_{i,j}^e \in \mathbb{R}^{D_e}$, respectively, where $D_v$ and $D_e$ stand for the dimensions of features of nodes and edges, respectively.

### Line graph transformer

To fully exploit the structural information from molecules, especially the chemical bonds that have been neglected in the previously defined transformer architectures[53–57], we transform the molecular graph $\mathcal{G} = (\mathcal{V}, \mathcal{E})$ to molecular line graph $\hat{\mathcal{G}} = \{\hat{\mathcal{V}}, \hat{\mathcal{E}}\}$ as the following two steps (Supplementary Fig. 3):

- For each edge $e_{i,j}$ in $\mathcal{G}$, create a node $\hat{v}_{i,j}$ in $\hat{\mathcal{G}}$;
- For every two edges in $\mathcal{G}$ that have a node in common, create an edge between their corresponding nodes in $\hat{\mathcal{G}}$.

We calculate the initial feature embedding $\boldsymbol{h}_{\hat{v}_{i,j}} \in \mathbb{R}^{D_{\hat{v}}}$ of node $\hat{v}_{i,j}$ in $\hat{\mathcal{G}}$ as follows:

$$\boldsymbol{h}_{\hat{v}_{i,j}} = \text{concat}(\boldsymbol{W}_v \boldsymbol{x}_i^v + \boldsymbol{W}_v \boldsymbol{x}_j^v, \boldsymbol{W}_e \boldsymbol{x}_{i,j}^e), \quad (1)$$

where $D_{\hat{v}}$ stands for the dimension of the initial feature embeddings of nodes in molecular line graphs, $\boldsymbol{W}_v \in \mathbb{R}^{\frac{D_{\hat{v}}}{2} \times D_v}$ and $\boldsymbol{W}_e \in \mathbb{R}^{\frac{D_{\hat{v}}}{2} \times D_e}$ stand

for the trainable projection matrices, and concat(·) stands for the concatenation operator. For clarity, we denote the nodes in a molecular line graph as $\hat{\mathcal{V}} = \{\hat{v}_i\}_{i \in [1, N_{\hat{v}}]}$ in the following sections, where $N_{\hat{v}}$ stands for the number of nodes in the molecular line graph.

We then propose Line Graph Transformer (LiGhT) to encode features of the molecular line graph. LiGhT is built upon a classic transformer encoder[50], which consists of multiple transformer layers. More specifically, given the node feature matrix $\boldsymbol{H} \in \mathbb{R}^{N_{\hat{v}} \times D_{\hat{v}}}$ of a molecular line graph, the transformer layer $l$ first feeds it into a multi-head self-attention module:

$$\boldsymbol{Q}^{l,k} = \boldsymbol{H}^{l-1} \boldsymbol{W}_Q^{l,k}, \boldsymbol{K}^{l,k} = \boldsymbol{H}^{l-1} \boldsymbol{W}_K^{l,k}, \boldsymbol{V}^{l,k} = \boldsymbol{H}^{l-1} \boldsymbol{W}_V^{l,k},$$

$$\boldsymbol{A}^{l,k} = \text{softmax}\left(\frac{\boldsymbol{Q}^{l,k}(\boldsymbol{K}^{l,k})^T}{\sqrt{D_h}}\right), \boldsymbol{H}^{l,k} = \boldsymbol{A}^{l,k} \boldsymbol{V}^{l,k}, \quad (2)$$

$$\boldsymbol{H}^l = \text{concat}(\boldsymbol{H}^{l,1}, \boldsymbol{H}^{l,2}, \dots, \boldsymbol{H}^{l,N_h}),$$

where $\boldsymbol{H}^{l-1}$ stands for the node feature matrix at the $(l-1)$-th layer, $\boldsymbol{W}_Q^{l,k} \in \mathbb{R}^{D_{\hat{v}} \times D_h}$, $\boldsymbol{W}_K^{l,k} \in \mathbb{R}^{D_{\hat{v}} \times D_h}$ and $\boldsymbol{W}_V^{l,k} \in \mathbb{R}^{D_{\hat{v}} \times D_h}$ stand for the trainable projection matrices of the $k$-th head at layer $l$, $D_h = \frac{D_{\hat{v}}}{N_h}$ stands for the dimension of each self-attention head, $N_h$ stands for the number of self-attention heads, and softmax(·) stands for the softmax operator. The output $\boldsymbol{H}^l$ is then passed to an FFN:

$$\hat{\boldsymbol{H}}^l = LN(\boldsymbol{H}^{l-1} + \boldsymbol{H}^l),$$

$$\boldsymbol{H}^l = LN\left(\boldsymbol{W}_2^l \text{GELU}(\boldsymbol{W}_1^l \hat{\boldsymbol{H}}^l) + \hat{\boldsymbol{H}}^l\right), \quad (3)$$

where LN(·) stands for the LayerNorm operator[87], GELU(·) stands for the GELU activation function[88], and $\boldsymbol{W}_1^l \in \mathbb{R}^{4D_{\hat{v}} \times D_{\hat{v}}}$ and $\boldsymbol{W}_2^l \in \mathbb{R}^{D_{\hat{v}} \times 4D_{\hat{v}}}$ stand for the trainable projection matrices at layer $l$.

Directly applying the above classic transformer architecture can lead to a significant loss of structural information of molecules since it ignores the connectivity of graphs. Therefore, we further employ path encoding and distance encoding modules to introduce the structural information into the multi-head self-attention layer.

**Path encoding module.** For each pair of nodes $\hat{v}_i$ and $\hat{v}_j$ in the molecular line graph, we first derive the shortest path between them and then encode the path features to an attention scalar $a_{i,j}^p$ in a path attention matrix $\boldsymbol{A}^p \in \mathbb{R}^{N_{\hat{v}} \times N_{\hat{v}}}$ as follows:

$$(\hat{v}_1^p, \hat{v}_2^p, \dots, \hat{v}_{N_p}^p) = \text{SP}(\hat{v}_i, \hat{v}_j),$$

$$a_{i,j}^p = \boldsymbol{W}_a^p \frac{1}{N_p} \sum_{n=1}^{N_p} \boldsymbol{W}_n^p \boldsymbol{h}_{v_n^p}, \quad (4)$$

where SP(·) stands for the shortest path function implemented by networkx[89], $(\hat{v}_1^p, \hat{v}_2^p, \dots, \hat{v}_{N_p}^p)$ stands for the shortest path between $\hat{v}_i$ and $\hat{v}_j$, $N_p$ stands for the length of the path, $\boldsymbol{h}_{v_n^p}$ stands for the feature of the $n$-th node in the shortest path, $\boldsymbol{W}_n^p \in \mathbb{R}^{D_p \times D_{\hat{v}}}$ stands for the trainable projection matrix for the $n$-th node in the path, $\boldsymbol{W}_a^p \in \mathbb{R}^{1 \times D_p}$ stands for a trainable projection matrix to project the path embedding to an attention scalar, and $D_p$ stands for the dimension of the path embedding.

**Distance encoding module.** Following refs. 54,57, we also leverage the distances between pairs of nodes to further encode the spatial features of the molecular line graphs. More specifically, given nodes $\hat{v}_i$ and $\hat{v}_j$ in a molecular line graph, we encode their distance to an attention scalar

$a_{i,j}^d$ in a distance attention matrix $\boldsymbol{A}^d \in \mathbb{R}^{N_{\hat{v}} \times N_{\hat{v}}}$ as follows:

$$
\begin{aligned}
d_{i,j} &= \text{SPD}(\hat{v}_i, \hat{v}_j), \\
a_{i,j}^d &= \boldsymbol{W}_2^d \text{GELU}(\boldsymbol{W}_1^d d_{i,j}),
\end{aligned}
\tag{5}
$$

where SPD($\cdot$) stands for the shortest path distance functoin, $d_{i,j}$ stands for the derived distance between $\hat{v}_i$ and $\hat{v}_j$, $\boldsymbol{W}_1^d \in \mathbb{R}^{D_d \times 1}$ and $\boldsymbol{W}_2^d \in \mathbb{R}^{1 \times D_d}$ stand for the trainable projection matrices, and $D_d$ stands for the dimension of the distance embedding.

Then, to introduce the encoded structural information into the model, we rewrite the formula of the attention matrix $\boldsymbol{A}^{l,k} \in \mathbb{R}^{N_{\hat{v}} \times N_{\hat{v}}}$ in the Eq. (2) as follows:

$$
\boldsymbol{A}^{l,k} = \text{softmax}\left( \frac{\boldsymbol{Q}^{l,k}(\boldsymbol{K}^{l,k})^T}{\sqrt{D_h}} + \boldsymbol{A}^p + \boldsymbol{A}^d \right).
\tag{6}
$$

where $\boldsymbol{A}^p$ and $\boldsymbol{A}^d$ are the path encoding matrix and the distance encoding matrix, respectively.

Here, we discuss the main advantages of our proposed model compared with the previously defined graph transformers:

First, by representing molecular graphs as line graphs, LiGhT emphasizes the importance of chemical bonds in molecules. Chemical bonds are the lasting attractions between atoms, which can be categorized into various types according to the ways they hold atoms together, resulting in different properties of the formed molecules. However, the previously defined transformer architectures either omit the edge features or only introduce chemical bonds as the bias in the self-attention module, ignoring the rich information from chemical bonds[53–57]. In our case, LiGhT fills this gap and fully exploits the intrinsic features of chemical bonds.

Second, although strategies like path encoding have already been proposed in previous graph transformer architectures[53,57] when encoding the paths, they only consider the edge features and ignore the node features in the paths. On the other hand, our path encoding strategy incorporates the features of the complete paths between pairs of nodes, thus encoding the structural information more precisely compared to the previous methods.

In summary, LiGhT provides a reliable backbone network for accurately modeling the structural and semantic information of molecular line graphs.

## The knowledge-guided pre-training strategy
**Knowledge.** In this study, we define knowledge as any quantifiable information that characterizes the features of molecules. This includes various types of information, such as molecular descriptors and fingerprints that are easily accessible through current cheminformatics tools[13,44]. Additionally, knowledge can encompass the experimentally measured characteristics of molecules, such as the comprehensive information on the bio-activities of 456,000 molecules across 1310 bioassays in the preprocessed ChEMBL dataset[90]. We employed 200 molecular descriptors and 512 RDKit fingerprints in our proposed method, which can be readily generated using RDKit[44], a widely used cheminformatics tool. The complete list of these molecular descriptors and examples of RDKit fingerprints can be found in Supplementary Table 14 and Supplementary Fig. 20, respectively.

**Pre-training strategy.** Our pre-training strategy is based on a generative self-supervised learning scheme, which first randomly selects a proportion of nodes in graphs. Then for each selected node, it is replaced with a mask token, a random node or the unchanged node with a ratio of 8:1:1. Finally, the model learns to predict the type of the original node with a cross-entropy loss. In the pre-training, we also randomly mask a proportion of the initial features of K nodes and learn to predict the masked molecular descriptors and fingerprints. The prediction of the masked molecular descriptor is formulated as a

regression task equipped with an RMSE loss, while the prediction of fingerprint is formulated as a binary classification task equipped with a cross-entropy loss.

## Finetuning strategies
To fully take advantage of the abundant knowledge captured in the pre-training stage, KPGT introduces four finetuning strategies, including layer-wise learning rate decay (LLRD)[61], re-initialization (ReInit)[61], FLAG[62] and, $L^2$-SP[63]. LLRD and ReInit are proposed mainly based on the fact that different layers of a model capture different kinds of information, where the bottom layers tend to encode the information more general to the downstream tasks while the top layers tend to encode information related to the pre-training tasks. More specifically, LLRD implements the discriminative learning rates for different layers of a model. This is achieved by setting an initial learning rate of the top layer and using a multiplicative decay rate to decrease the initial learning rate layer-by-layer from top to bottom. ReInit re-initializes the parameters of the top-$n$ layers of a model before finetuning. FLAG is a data augmentation method that iteratively augments the node features by injecting the gradient-based adversarial perturbations during finetuning. $L^2$-SP proposes a regularization scheme to explicitly promote the similarity of the finetuned model with the initial one in the finetuning process.

## Training details
All models were implemented in PyTorch[91] version 1.10.0 and DGL[92] version 0.7.2 with CUDA version 11.3 and Python 3.7. We implemented a 12-layer LiGhT as the backbone network with 768 hidden units and 12 self-attention heads. A mean pooling operation that averaged all the nodes in individual graphs was applied on top of the model to extract the molecular feature representations. An Adam optimizer with weight decay $1e^{-6}$ and learning rate $2e^{-4}$ was used to optimize the model. The model was trained with a batch size of 1024 for a total of 100,000 steps. The KPGT had around 100 million parameters. We set the masking rate of both nodes and additional knowledge to 0.5. The pre-training of KPGT took about two days on four Nvidia A100 GPUs. More detailed configurations of KPGT in the pre-training and finetuning processes are summarized in Supplementary Table 3.

## Statistics and reproducibility
In our first benchmarking test in the "Results" section, we employed 11 molecular property datasets, including eight classification datasets and three regression datasets. Detailed information of these datasets is available in Supplementary Note 1.1. For this evaluation test, following the established practice from previous research[6,38], a scaffold splitting scheme was utilized to partition each dataset into training, validation, and test sets with a ratio of 8:1:1. This splitting scheme ensured the molecules in the test sets differed structurally from those in the training set, offering an ideal scenario to evaluate the robustness and generalizability of prediction models. We utilized the area under the receiver operating characteristic curve (AUROC) to evaluate classification tasks, while for regression tasks, we employed the root-mean-square error (RMSE) as the evaluation metric. The results were reported based on three independent runs with different random seeds.

The TDC benchmarking platform provides 22 molecular property prediction datasets. These datasets span a broad spectrum of molecular properties, including absorption, distribution, metabolism, excretion, and toxicity (ADMET). We strictly adhered to the evaluation protocols provided by the original TDC benchmarking platform. Each dataset was split into training, validation, and test sets with a ratio of 7:1:2 using a scaffold splitting scheme. The mean absolute error (MAE) or Spearman's rank correlation coefficient (Spearman's r) was employed to evaluate the prediction performance of KPGT on

regression tasks, while AUROC or the area under the precision-recall curve (AUPRC) was used to evaluate the prediction performance of KPGT on classification tasks (the specific metric used for each dataset can be found in Supplementary Table 8). The results were reported based on five independent runs with different random seeds.

MoleculeACE provides datasets that measure the bio-activities of molecules against 30 macromolecular targets. We followed the same evaluation protocols as provided by the benchmark. For each dataset from the benchmarking platform, molecules were first clustered based on their extended-connectivity fingerprints (ECFPs)[14] using spectral clustering[93]. Then for each cluster, molecules were split into training and test sets with a ratio of 8:2 using stratified random sampling based on their activity cliff labels[66].

For the tests on the datasets for HPK1, we evaluated the prediction performance of KPGT on this dataset in three distinct scenarios: (1) scaffold splitting, i.e., splitting the dataset as training, validation, and test sets with a ratio of 8:1:1 using scaffold splitting; (2) time splitting, i.e., splitting the dataset according to the time that the corresponding patents were published (the molecules from the patents published before 2021 were used as training while those molecules published after 2021 were used as testing); and (3) domain transfer, i.e., collecting 1615 molecules with the negative logarithm of inhibitory constants (pKi) against HPK1 as an independent test set. We excluded the molecules that overlapped with the pre-training data employed by KPGT in this test. Spearman's r and Pearson correlation coefficient (Pearson's r) were employed as the evaluation metrics. The results were reported based on three independent runs with different random seeds.

For the tests on the datasets for FGFR1, we evaluated the prediction performance of KPGT in the scaffold splitting and time splitting scenarios. We excluded the molecules that overlapped with the pre-training data employed by KPGT in this test. Spearman's r and Pearson's r were employed as the evaluation metrics. The results were reported based on three independent runs with different random seeds.

### Reporting summary
Further information on research design is available in the Nature Portfolio Reporting Summary linked to this article.

## Data availability
All relevant data supporting the key findings of this study are available within the article and its Supplementary Information files. The processed ChEMBL29 dataset and the datasets from the first benchmarking tests in the "Results" section can be accessed via Figshare (https://doi.org/10.6084/m9.figshare.19914811). The datasets from the TDC benchmark platform are available at https://tdcommons.ai/benchmark/overview/. The datasets from the MoleculeACE benchmark are available at https://github.com/molML/MoleculeACE. The HPK1 and FGFR1 datasets can be accessed via Figshare (https://doi.org/10.6084/m9.figshare.24290899). The FDA dataset is available at https://go.drugbank.com/releases/5-1-10/downloads/approved-structure-links. The reference protein-ligand complex structures for HPK1 and FGFR1 used in this study are available in the Protein Data Bank under accession codes 7SIU [https://www.rcsb.org/structure/7SIU] and 5A4C [https://www.rcsb.org/structure/5A4C], respectively. Preliminary results from this study have been reported in the conference proceedings of ref. 58. Source data are provided with this paper.

## Code availability
The source code of KPGT can be downloaded from the GitHub repository at https://github.com/lihan97/KPGT or the Zenodo repository at https://doi.org/10.5281/zenodo.8418818[94].

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

## Acknowledgements

This work was supported in part by the National Natural Science Foundation of China (T2125007 to J.Z. and 32270640 to D.Z.), the National Key Research and Development Program of China (2021YFF1201300 to J.Z.), the New Cornerstone Science Foundation through the XPLORER PRIZE (J.Z.), the Research Center for Industries of the Future (RCIF) at Westlake University (J.Z.) and the Westlake Education Foundation (J.Z.).

## Author contributions

H.L., D.Z., and J.Z. conceived the concept. H.L., R.Z., and Y.M. designed the methodology and performed computational experiments. H.L., R.Z., Y.M., D.M., D.Z., and J.Z. analyzed the results. H.L., R.Z., and D.M. wrote the paper with the help of all authors.

## Competing interests

J.Z. is the founder of Silexon AI Technology Co., Ltd. and has an equity interest. All other authors declare no competing interests.
