## [Peer Review File · Nature Communications]

Reviewers' Comments:

Reviewer #1:

Remarks to the Author:

The study highlights the importance of effective molecular feature representation in drug discovery. The researchers propose a novel framework called Knowledge-guided Pre-training of Graph Transformer (KPGT), which leverages unlabeled molecules to learn generalizable and robust molecular representations by integrating a Line graph transformer (LiGhT) and a knowledge-guided pre-training strategy to capture both structural and semantic knowledge of molecules. Specifically, LiGhT utilizes node embeddings, distance encoding, and path encoding in a transformer encoder for effective molecule representation. Knowledge is used to transform molecule graphs into molecular line graphs, providing semantic information about molecules. This enables KPGT to contain more information than just structures. However, this paper is missing an important discussion and experiments about previous work, and as a result, this work could be rejected. The following are specific comments or suggestions:

1. Ablation studies on pre-training and no pre-training of KPGT are presented in Supplementary Figure S10, and a comparison with other backbone models is shown in Supplementary Figure S13. Here are a few suggested corrections and improvements to the provided text:

- Since this paper introduces the innovative concept of knowledge-based molecular line graphs for semantic information, the ablation studies should also include an analysis of the replacement of the molecular line graph. For instance, one ablation study could focus on the node feature in Graphormer.

- In my opinion, the backbone models of KPGT are similar to Graphormer. However, a distinction lies in the features used for input and the tasks used for pre-training. In this case, the effectiveness of this modification should be clarified through experimentation. For example, consider evaluating a variant of KPGT without distance encoding or path encoding.

Please note that I have made some assumptions and rephrased parts of the text for clarity. Feel free to adjust the suggestions according to the context and your specific requirements.

2. With a molecular line graph generated by knowledge, knowledge prediction, and masked node prediction, the representation of the molecule is enriched with semantic and structural information. Figure 3d demonstrates that KPGT can effectively distinguish between inhibitor and non-inhibitor molecules, indicating that KPGT has learned the semantic differences between these two types of molecules. However, it is important to also display the distribution of inhibitor and non-inhibitor molecules for the baselines.

3. In the baseline experiment, the methods of masking, contextpred, maskingsup, and contextpredsup belong to the same pre-training strategy called "STRATEGIES FOR PRE-TRAINING GRAPH NEURAL NETWORKS." In addition to these pre-training tasks, pre-training gnn also includes combinations such as Supervised+ContextPred, Supervised+AttrMasking, and Supervised+EdgePred. Among these combinations, the one combining Supervised+ContextPred demonstrates the best performance. Therefore, it is essential to compare the results of our method with this particular combination.

4. I am curious to know whether cross-validation was used in the results presented in Figures 2(a) and 2(b). If cross-validation was employed, it is necessary to provide details about the specific cross-validation procedure utilized. On the other hand, if cross-validation was not utilized, it is recommended to incorporate cross-validation into the analysis.

5. Figure 4 illustrates the identification of HPK1 inhibitors by KPGT. To enhance the presentation, it is suggested to include the docking score of the molecule identified by KPGT.

6. Figure 2(d) displays the RMSE score of MoleculeACEALL and MoleculeACEcliff under KPGT, SVM, GBM, and RF. It is worth noting that while SVM, GBM, and RF are machine learning methods, KPGT is a deep neural network. To ensure a comprehensive comparison, the experiments involving MoleculeACEALL and MoleculeACEcliff should encompass the deep learning method, and the machine learning method should be added to the Supplementary file.

7. To the best of my understanding, the knowledge utilized in this paper refers to molecular descriptors and fingerprints. However, there seems to be an issue regarding the definitions of these molecular descriptors and fingerprints. According to lines 156-157, the definitions should be presented in Notes 1.2. However, upon reviewing S1.2 in the Supplementary file, it appears that it provides only a brief introduction to the baseline methods instead. It is crucial to accurately describe the definitions of molecular descriptors and fingerprints in the appropriate section.

8. Lines 186-188 indicate that additional knowledge is utilized in the self-supervised learning methods Contextpredsup, Maskingsup, and GROVER. However, it is unclear what specific types of knowledge are employed in these methods. Are you referring to context, masked nodes or edges, or other forms of knowledge? It is important to note that the author has previously defined molecule descriptors and fingerprints as knowledge. Masked nodes, edges, context, and properties cannot be classified as knowledge in the same way. If you wish to include these elements as part of the knowledge in this paper, it is necessary to clearly define the scope of knowledge within the context of this study.
9. Overall, this paper is readable; however, there are issues with the logic and framework as they are not well structured.
10. There are also some grammar errors in this article.

Reviewer #2:

Remarks to the Author:

This paper proposed a novel knowledge-guided pre-training of graph transformer, named (KPGT), for learning molecular representations. KPGT combines a graph transformer designed for molecular graphs with a knowledge-guided pre-training strategy to capture molecule knowledge. The authors evaluate their method on various datasets and compare it with several baseline methods. The results demonstrated the superior performance of KPGT on predicting molecular properties. Overall, the paper is interesting and well-presented. However, the novelty appears a bit weak as some self-supervised pre-training frameworks similar to KPGT have been previously developed for molecule representation learning, such as Fang et al., 2022, Xu et al., 2021, and Rong et al, 2020 (see below for the reference). There are a few concerns that should be addressed to strengthen the study.

1. Clarify Splitting Schemes: The authors use different splitting schemes for different tasks, which may lead to confusion. For example, the authors divide the dataset into training, validation, and test sets with an 8:1:1 ratio in the classification task, while it is divided with a 7:1:2 ratio in the molecular property prediction tasks. It would be helpful to provide a clear explanation of the rationale behind these choices and clarify whether the proposed model is robust against different splitting schemes.
2. Include Baseline Results: The results of many baselines are missing from the manuscript. For example, in section 2.2, the authors claim that they compare KPGT with 17 baseline methods on 22 molecular property prediction tasks from the TDC benchmark, while with 24 baseline methods on the datasets from the MoleculeACE benchmark. Unfortunately, the authors only show the results of the best baseline methods evaluated by the benchmarking papers. It would be beneficial to include the results of all evaluated baseline methods for a more comprehensive comparison.
3. Compare with Advanced Models: The authors claim that "These results suggested that the previously designed deep learning models might fail to capture the intrinsic features of molecules". It seems that most of baseline methods used in the benchmarking papers are traditional methods. Recently, more advanced self-supervised models for molecular property prediction have been proposed, such as Xu et al., 2021 and Fang et al. 2022. To support this claim and further demonstrate the superiority of KPGT, it would be valuable to compare it with more advanced self-supervised models.
4. Provide Explanation for Figure 2c: "As shown in Figure 2c, KPGT achieved the best prediction performance on sixteen out of ...". but it is unclear how this conclusion is made. Additional explanations regarding Figure 2c would help readers understand the authors' findings.
5. Justify Target Selection: In section 2.4, the authors take two antitumor targets HPK1 and FGFR1 as examples to evaluate the practical applicability of KPGT. It would be helpful to provide justification for choosing these targets and discuss whether the proposed model can be generalized to other targets.
6. Conduct Ablation Study: In the proposed framework, the authors introduce path encoding and distance encoding modules, which, I think, are critical components of the proposed model. Conducting an ablation study to test the contributions of these modules to the model's performance would provide a deeper understanding of their effectiveness.
7. Add a Discussion Section: It is recommended adding a dedicated 'Discussion' section to the manuscript. This section would allow the authors to discuss the advantages and limitations of the proposed method, as well as its potential applications in practical biological scenarios.

8. Justify "Knowledge-Guided": The authors describe their method as "knowledge-guided" only because they use prior knowledge as node features. However, it might be misleading as the utilization of prior knowledge as node features is relatively common. To justify the use of the term "knowledge-guided", the authors could provide more compelling reasons or evidence for why their use of prior knowledge goes beyond common practices in the field.
9. Address Pre-training Time Concerns: It is mentioned that the pre-training progress takes a long time, i.e., about 2 days. This extended duration may pose a challenge for non-experts or users with limited computational resources. It would be beneficial to address this concern by discussing potential strategies for reducing the pre-training time or providing guidance on how to handle such computational requirements.
10. Provide Supporting Evidence: "Such backbone models generally provide low model capacity and thus potentially fail to capture the wide range of information required for the prediction of various properties for a diversity of molecules." The description might not be accurate. To support this claim, it would be advantageous to cite relevant literature that demonstrates the limitations of previous models in capturing diverse molecular properties.
11. Update Figure 1: The legend of Figure 1 mentions abbreviations, but the DE and PE modules are not reflected in the figure. This inconsistency should be addressed to ensure clarity.
12. Make Code Accessible: Although the authors provide a code link, it is currently inaccessible, making it difficult to validate the reproducibility of the reported results. The authors are suggested to make the code publicly accessible.

Reference

- [1]. Fang et al. Geometry-enhanced molecular representation learning for property prediction. Nature Machine Intelligence. 2022.
- [2]. Xu et al. Self-supervised Graph-level Representation Learning with Local and Global Structure. ICML. 2021.
- [3]. Rong et al. Self-Supervised Graph Transformer on Large-Scale Molecular Data. NeurIPS. 2020.

Reviewer #3:

Remarks to the Author:

In this paper, Li and others proposed KPGT, a novel knowledge-guided transformer model. This method is an integration of graph transformer and a pre-training strategy. It has been used for molecular property prediction (compared to 55 baselines on 63 datasets), and identify anti-tumor inhibitors. Overall, KPGT presents a novel pre-training framework, which could be a game changer for drug discovery. The extensive results also confirm the effectiveness of KPGT. I have a few comments for this manuscript to further improve and clarify its contribution.

1. Section 2.2 "The mean absolute error (MAE) and Spearman's correlation coefficient (Spearman's r) were employed to evaluate the prediction performance of KPGT on regression tasks, while AUROC and the area under the precision-recall curve (AUPRC) were used to evaluate the prediction performance of KPGT on classification tasks", where are the results of AUPRC and correlation? I didn't find it in the main figures or supplementary figures.
2. Please release the code and data for the 55 baselines and how to reproduce their results from the 63 datasets.
3. please clarify why most of the baselines used in figure 2 are not compared in the experiment in figure 4. Are they not applicable in the time splitting and domain transfer settings?
4. the drug molecular graphs showed in figure 4e are not informative. Do drug structures reflect potency?
5. It might be interesting to perform graph-based explanation approach (GNExplainer) to understand the important atoms and sub-structures in each graph in figure 4e.
6. For the 4442 molecules identified for validating HPK1, are they excluded from the pre-trained data used by KPGT?

How We Addressed the Reviewers' Comments for "A Knowledge-Guided Pre-training Framework for Improving Molecular Representation Learning"

Han Li¹, Ruotian Zhang¹, Yaosen Min¹, Dacheng Ma², Dan Zhao^{1,*}, and Jianyang Zeng^{1,3,*}

¹ Institute for Interdisciplinary Information Sciences, Tsinghua University, 100084, Beijing, China.

² Research Center for Biological Computation, Zhejiang Laboratory, 311100, Hangzhou, China.

³ Present address: School of Engineering, Westlake University, Zhejiang Province, 310030, Hangzhou, China.

* Corresponding authors: Dan Zhao, E-mail: zhaodan2018@tsinghua.edu.cn, and Jianyang Zeng, E-mail: zengjy@westlake.edu.cn.

We thank the editors and reviewers for the insightful and valuable comments and suggestions, which are helpful in improving the quality of our manuscript. We have revised our manuscript according to these comments and suggestions. In this response, we provide point-by-point answers to the reviewers' comments. In particular, we have improved our manuscript through addressing the following major concerns raised by the reviewers:

1. We have conducted additional tests to further assess the applicability and reliability of KPGT in predicting molecular properties. More specifically, we have included five new advanced self-supervised learning-based baseline methods in our evaluation tests to thoroughly compare and validate the superiority of KPGT. Furthermore, we have included nineteen self-supervised learning methods in the tests conducted on the datasets from the MoleculeACE benchmarking platform to provide a clearer and more comprehensive conclusion. We have further carried out docking analyses on the molecules identified by KPGT that potentially inhibit antitumor targets to strengthen our findings. In addition, we have conducted ablation studies to individually validate the effectiveness of the specific design components within KPGT.
2. We have carefully revised our manuscript to make our paper more readable and easier to understand. In particular, we have provided a detailed exploration of the evolution of AI-based approaches for molecular property prediction in the Introduction section. Moreover, we have clarified several key concepts used in our manuscript, such as knowledge, molecular descriptors, and fingerprints, ensuring that readers can easily grasp the essential ideas. We have also improved the descriptions of our experimental settings, providing clearer explanations about the baseline methods and data splitting details. To provide a better summary of our work, we have also added a Discussion section that discusses the advantages, limitations, and potential applications of our proposed method. In order to present our framework more coherently, we have reorganized sections and subsections, creating a smoother flow of information. Lastly, we have carefully proofread the entire manuscript to guarantee the accuracy and clarity of the language used.
3. To ensure that our work can be easily reproduced and utilized by the research community, we have taken steps to make our source code and pre-trained model available on GitHub, which can be accessed through the following link: <https://github.com/lihan97/KPGT>. To further assist users in applying our pre-trained model to their specific molecular property datasets, we have also created a user-friendly README file containing detailed instructions for the finetuning process.

We believe that our revised manuscript should have addressed all the reviewers' concerns. In the revised manuscript, the corresponding modifications have been highlighted in red. In the remaining part of this response, we will provide point-by-point responses to the reviewers' comments.

Reviewer 1

1. Remarks to the Author: The study highlights the importance of effective molecular feature representation in drug discovery. The researchers propose a novel framework called Knowledge-guided Pretraining of Graph Transformer (KPGT), which leverages unlabeled molecules to learn generalizable and robust molecular representations by integrating a Line graph transformer (LiGhT) and a knowledge-guided pre-training strategy to capture both structural and semantic knowledge of molecules. Specifically, LiGhT utilizes node embeddings, distance encoding, and path encoding in a transformer encoder for effective molecule representation. Knowledge is used to transform molecule graphs into molecular line graphs, providing semantic information about molecules. This enables KPGT to contain more information than just structures. However, this paper is missing an important discussion and experiments about previous work, and as a result, this work could be rejected. The following are specific comments or suggestions:

Response: We thank the reviewer for the nice summary of our work. We acknowledge the reviewer's concern that "this paper is missing an important discussion and experiments on previous works". We agree with the reviewer that it is crucial to provide a thorough discussion of previous works and conduct comprehensive experiments to precisely assess the contribution of our work. To address the specific concern of "missing an important discussion" raised by the reviewer, we have thoroughly revised the Introduction section of our manuscript to encompass a more systematic exploration of the landscape regarding previous works for molecular property prediction. To elaborate, we have sequentially introduced various stages in the evolution of artificial intelligence (AI)-based approaches within this domain. Our revised Introduction section commences with an overview of early machine learning-based approaches, which represent molecules using basic handcrafted features¹⁻³. We also highlighted the limitations of these methods, which heavily rely on intricate feature engineering strategies, resulting in a lack of generalizability and flexibility. Subsequently, we delved into the emergence of deep learning-based techniques, including recurrent neural networks (RNNs)⁴⁻⁶, convolutional neural networks (CNNs)⁷⁻⁹, and graph neural networks (GNNs)¹⁰⁻¹³. We elucidated their strengths while also acknowledging their inherent limitations, such as suffering from data scarcity issues and an inability to generalize effectively to out-of-distribution data samples. Furthermore, we navigated through the landscape of current self-supervised learning-based methods¹⁴⁻²³, shedding light on their contributions and pointing out their primary limitations, that is, the ill-defined pre-training strategy and limited capacity of backbone models. These discussions holistically encapsulate the evolutionary journey of AI-based methodologies for molecular property prediction. These discussions also serve as a potent motivation for the introduction of our own work. We provide the relevant paragraphs from the revised Introduction section (Lines 29-77, Page 2) below for review:

"In recent years, artificial intelligence (AI)-based approaches have played an increasingly pivotal role in measuring molecular properties, offering remarkable efficiency and cost-effectiveness³⁻⁵. One of the primary challenges of AI-based approaches for molecular property prediction is the efficient representation of molecules^{6,7}. The early machine learning approaches involved preliminary attempts towards representing molecules using basic handcrafted features⁸⁻¹⁰ ... The prediction methods based on these representations are highly dependent on complicated feature engineering strategies, consequently compromising their generalizability and flexibility. ... Notably, recurrent neural networks (RNNs)¹⁷⁻¹⁹, convolutional neural networks (CNNs)²⁰⁻²², and graph neural networks (GNNs)²³⁻²⁶ have demonstrated their exceptional ability to model molecular data in various formats ... Nevertheless, the limited availability of labeled molecules and

the vastness of the chemical space have constrained their prediction performance, particularly when handling out-of-distribution data samples^{6,28,29}. Along with the remarkable achievements of self-supervised learning methods in the fields of natural language processing^{30,31} and computer vision^{32,33}, these techniques have been employed to pre-train GNNs and improve the representation learning of molecules, leading to substantial improvements in the downstream molecular property prediction tasks^{28,34-42}.

Current self-supervised learning methods, exemplified by GraphLoG³⁶, GROVER³⁸, and GEM42, typically involve the modification of molecular graphs by means of node or subgraph masking, followed by the prediction of the masked components^{28,38}, or the utilization of contrastive learning objectives to align the modified graphs with their corresponding originals in latent space^{35,37} ... This natural characteristic of molecules potentially limits current self-supervised learning-based methods for molecular graphs to capture structural similarity among molecules and fails to capture the rich semantic information related to molecular properties encoded within their chemical structures (Supplementary Fig. 1)⁴³ ...

Existing self-supervised learning methods generally rely on GNNs (e.g., graph isomorphism network⁴⁵) as backbone models. However, GNNs can only provide limited model capacity, as they suffer from over-smoothing when increasing their number of layers^{46,47} ... These models, characterized by an increasing number of parameters and the capability to capture long-range interactions, present promising avenues to comprehensively model the structural characteristics of molecules⁵³⁻⁵⁷.

In response to the reviewer's concern regarding the missing experiments on previous works, we have conducted additional tests to improve the comprehensiveness of our evaluation. Precisely, within our first benchmarking test in the Results section, we have extended our evaluation to include five advanced self-supervised learning-based methods, including GEM²³, GraphMAE²⁴, MoleBERT²⁵, Supervised+Edgepred (Edgepred_{Sup})¹⁴, and Supervised+Infomax (Infomax_{Sup})¹⁴. In the feature extraction setting, KPGT exhibited superior performance compared to baseline methods across seven out of eight classification datasets and two out of three regression datasets, leading to overall relative improvements of 2.0% for classification and 4.5% for regression (Fig. R1a and Supplementary Tables 4-5). In the finetuning setting, KPGT outperformed baseline methods across seven out of eight classification datasets and all three regression datasets, resulting in overall relative improvements of 1.6% for classification and 4.2% for regression (Fig. R1b and Supplementary Tables 6-7). Additionally, we have conducted a further evaluation by comparing KPGT with the nineteen self-supervised learning-based methods employed in the first benchmarking test on the datasets provided by MoleculeACE²⁶. Fig. R2a and Table R2 report the results of KPGT and baseline methods. The results demonstrated that KPGT outperformed baseline methods on twenty-six out of thirty datasets, with an overall relative improvement of 3.9%. Fig. R2b and Table R3 present the results of KPGT and baseline methods evaluated exclusively on the activity cliffs within each test set. From the results, KPGT achieved superior prediction performance in comparison with baseline methods on twenty-two out of thirty datasets, exhibiting an overall relative improvement of 1.2%. Furthermore, to thoroughly evaluate the prediction performance of KPGT on identifying inhibitors for the specific antitumor targets, namely HPK1 and FGFR1, we have conducted supplementary tests incorporating all the self-supervised learning-based baseline methods used in the first benchmarking test in the Results section. As shown in Figs. R3-R4, KPGT consistently outperformed all the baseline methods on the HPK1 and FGFR1 datasets in different evaluation scenarios. Collectively, these additional evaluation results enabled us to present a more comprehensive and detailed comparison between KPGT and prior works in predicting molecular properties. By revising our Introduction section and conducting these sup-

plementary evaluations, we believe that we have effectively addressed the reviewer's concerns and further enhanced the rigor of our manuscript.

We have carefully considered the reviewer's comments and taken them into account to strengthen the quality and impact of our study. Below, we will provide a detailed response addressing each of the reviewer's comments.

Fig. R1: Benchmarking the performance of KPGT against nineteen self-supervised learning-based baseline methods. **a-b** The averaged prediction performance of KPGT and the self-supervised learning-based baseline methods for classification (measured by AUROC) and regression datasets (measured by RMSE) in two settings: (a) feature extraction, where the pre-trained LiGhT model remains fixed, and (b) transfer learning, where the pre-trained LiGhT model is trainable. These results were reported based on three independent runs with different random seeds.

2. Ablation studies on pre-training and no pre-training of KPGT are presented in Supplementary Figure S10, and a comparison with other backbone models is shown in Supplementary Figure S13. Here are a few suggested corrections and improvements to the provided text:

- Since this paper introduces the innovative concept of knowledge-based molecular line graphs for semantic information, the ablation studies should also include an analysis of the replacement of the molecular line graph. For instance, one ablation study could focus on the node feature in Graphormer.
- In my opinion, the backbone models of KPGT are similar to Graphormer. However, a distinction lies in the features used for input and the tasks used for pre-training. In this case, the effectiveness of this modification should be clarified through experimentation. For example, consider evaluating a variant of KPGT without distance encoding or path encoding.

Response: We appreciate the reviewer for the insightful suggestion regarding the necessity of conducting ablation studies on KPGT. These ablation studies allow us to thoroughly understand the individual contributions of specific design components within our proposed method, such as

Fig. R2: Averaged prediction performance of KPGT and baseline methods on the datasets from the MoleculeACE benchmark under the (a) MoleculeACE_{All} and (b) MoleculeACE_{Cliff} settings, respectively, measured in terms of RMSE.

the incorporation of the molecular line graph as well as the distance encoding and path encoding modules integrated into LiGhT. In response to this, we have conducted more comprehensive ablation studies on KPGT. To elaborate, we introduced several modified frameworks based on KPGT with specific restrictions: KPGT-KN (without knowledge node), KPGT-PE (without path encoding module), KPGT-DE (without distance encoding module), KPGT-LG (replacing molecular line graph with the molecular graph used in Graphormer²⁷), KPGT-LiGhT+Graphormer (replacing backbone model LiGhT with Graphormer²⁷), and KPGT-LiGhT+GIN (replacing backbone model LiGhT with GIN²⁸). Table R1 reports the performance of KPGT and modified frameworks on the datasets from our first benchmarking test in the Results section. The results showcased the superiority of KPGT over KPGT-LG, resulting in an overall relative improvement of 2.3%. This observation demonstrated the enhanced informativeness of the molecular line graph utilized in KPGT, in contrast to the original molecular graph in Graphormer²⁷. KPGT also outperformed KPGT-LiGhT+Graphormer with an overall relative improvement of 1.8%, indicating the enhanced capacity of our proposed backbone model, LiGhT, in effectively capturing the inherent structural information in comparison with Graphormer. Moreover, KPGT surpassed KPGT-PE and KPGT-DE with overall relative improvements of 2.7% and 3.0%, respectively, providing empirical validation for the pivotal role played by the distance encoding and path encoding modules. Overall, KPGT outperformed all the modified frameworks, highlighting the significant contributions of the individual design components of KPGT to its superior performance. In our revised manuscript, we have updated the relevant texts in the subsection named "Ablation Studies on KPGT" in the Results section (Lines 308-325, Page 13) and included these results in Supplementary Table 13. We present the revised subsection in the Results section below for review:

"To validate the effectiveness of the specific design choices of KPGT, we conducted comprehensive ablation studies. Specifically, we introduced several modified frameworks based on KPGT with specific restrictions: KPGT-Pretrain (without pre-training), KPGT-KN (without knowledge nodes),

Fig. R3: Prediction performance of KPGT and baseline methods on the HPK1 dataset under the scaffold splitting, time splitting, and domain transfer scenarios, measured in terms of both Spearman's r and Pearson's r .

Fig. R4: Prediction performance of KPGT and baseline methods on the FGFR1 dataset under the scaffold splitting and time splitting scenarios, measured in terms of Spearman's r and Pearson's r .

KPGT-PE (without path encoding module), KPGT-DE (without distance encoding module), KPGT-LG (replacing molecular line graph with original molecular graph), KPGT-LiGhT+Graphormer (replacing backbone model LiGhT with Graphormer⁵⁷), and KPGT-LiGhT+GIN (replacing backbone model LiGhT with GIN⁴⁵). Additionally, for a fair comparison, we ensured that all models had approximately the same number of parameters (around 3.5 million). Supplementary Table 13 reports the performance of KPGT and modified frameworks on the datasets from our first benchmarking test in the Results section. The results showcased the superiority of KPGT over KPGT-LG, resulting in an overall relative improvement of 2.3%. This observation demonstrated the enhanced informativeness of the molecular line graph utilized in KPGT, in contrast to the original molecular graph in Graphormer⁵⁷. KPGT also outperforms KPGT-LiGhT+Graphormer with an overall relative improvement of 1.8%, indicating the enhanced capacity of our proposed backbone model, LiGhT, in effectively capturing the inherent structural information in comparison with Graphormer. Moreover, KPGT surpassed KPGT-PE and KPGT-DE with overall relative improvements of 2.7% and 3.0%, respectively, providing empirical validation for the significant role played by the distance encoding and path encoding modules. Overall, KPGT outperformed all the modified frameworks, highlighting the significant contributions of the individual design components of KPGT to its superior performance.”

Moreover, as commented by the reviewer, the backbone of KPGT, LiGhT, is similar to Graphormer²⁷. Both models leverage the classic transformer architecture²⁹ and incorporate positional encoding modules specifically designed for handling molecular graphs. The key factors distinguishing LiGhT and Graphormer are the incorporation of molecular line graphs and an improved path encoding module. Through representing molecules as line graphs, LiGhT can fully capture the inherent feature of chemical bonds, a facet that has been somewhat overlooked in previously proposed graph transformer architectures, including Graphormer²⁷. Furthermore, although Graphormer²⁷ also employs a path encoding module, it only takes into account the features of chemical bonds, disregarding atom-related features during the path encoding process. The application of the line graph representation empowers the path encoding module within LiGhT to capture both chemical bond and atom features in their entirety. These advancements equip LiGhT with the capability to precisely capture the intrinsic structural information inherent in molecules.

3. With a molecular line graph generated by knowledge, knowledge prediction, and masked node prediction, the representation of the molecule is enriched with semantic and structural information. Figure 3d demonstrates that KPGT can effectively distinguish between inhibitor and non-inhibitor molecules, indicating that KPGT has learned the semantic differences between these two types of molecules. However, it is important to also display the distribution of inhibitor and non-inhibitor molecules for the baselines.

Response: We appreciate the reviewer for the valuable suggestion. According to the reviewer’s comments, while Fig. 3d showcases the capability of KPGT to distinguish between inhibitors and non-inhibitors in the latent space, it is essential to include the results of baseline methods in order to substantiate our conclusion. Following the reviewer’s suggestion, we have incorporated additional visualization tests to enhance the clarity of our findings. Specifically, we have employed t-distributed stochastic neighbor embedding (t-SNE)³⁰ to visualize the distributions of inhibitors and non-inhibitors from the test set of the CYP3A4 dataset³¹ for the baseline methods, namely ECFP³², RDKFP³³, GROVER¹⁹, and GraphCL¹⁶. Our analysis revealed that all the baseline methods failed to provide a clear separation between inhibitors and non-inhibitors in the latent space

Table R1: Ablation studies of KPGT. Several modified frameworks based on KPGT with specific restrictions are introduced, including KPGT-Pretrain (without pre-training), KPGT-KN (without the knowledge nodes), KPGT-PE (without the path encoding module), KPGT-DE (without the distance encoding module), KPGT-LG (replacing molecular line graph with the molecular graph used in Graphormer), KPGT-LiGhT+Graphormer (replacing the backbone model LiGhT with Graphormer), and KPGT-LiGhT+GIN (replacing the backbone model LiGhT with GIN). The averaged results on the classification and regression datasets from the first benchmarking test in the Results section were reported, measured in terms of AUROC and RMSE, respectively. The numbers in brackets are the standard deviations across three independent runs. The best results are marked in bold.

Method	Classification dataset	Regression dataset
	AVG (AUROC)	AVG (RMSE)
KPGT-LiGhT+GIN	0.817 _(0.009)	1.229 _(0.257)
KPGT-LiGhT+Graphormer	0.822 _(0.007)	1.260 _(0.081)
KPGT-LG	0.824 _(0.006)	1.250 _(0.303)
KPGT-PE	0.822 _(0.007)	1.281 _(0.254)
KPGT-DE	0.823 _(0.012)	1.306 _(0.144)
KPGT-KN	0.819 _(0.008)	1.287 _(0.304)
KPGT-Pretrain	0.792 _(0.001)	1.469 _(0.302)
KPGT	0.829 _(0.011)	1.162 _(0.197)

(Fig. R5). Moreover, we have assessed the prediction performance of KPGT and baseline methods in classifying activity cliffs within the CYP3A4 dataset. As shown in Fig. R6, KPGT achieved an overall relative improvement of 7.1% in comparison with baseline methods. These results highlighted the superior sensitivity of KPGT in capturing the semantic changes of molecules, even in cases where such changes arose from subtle structural alterations. In the revised manuscript, we have incorporated these visualization results into Supplementary Fig. 11 and refined the prediction results presented in Supplementary Fig. 12.

- In the baseline experiment, the methods of masking, contextpred, maskingsup, and contextpred-sup belong to the same pre-training strategy called "STRATEGIES FOR PRETRAINING GRAPH NEURAL NETWORKS." In addition to these pre-training tasks, pretraining GNN also includes combinations such as Supervised+ContextPred, Supervised+AttrMasking, and Supervised+EdgePred. Among these combinations, the one combining Supervised+ContextPred demonstrates the best performance. Therefore, it is essential to compare the results of our method with this particular combination.

Response: We apologize for any confusion regarding the definition of the baseline methods employed. As detailed in the subsection titled "KPGT outperforms the baseline methods on molecular property prediction" within the Results section of our original manuscript, we utilized four pioneering self-supervised learning-based methods introduced by Hu et al.¹⁴. These methods were: Attribute Masking (abbreviated as Masking), Context Prediction (abbreviated as Contextpred), Supervised+Masking (abbreviated as Masking_{sup}), and Supervised+Context Prediction (abbrevi-

Fig. R5: Visualization of the distributions of inhibitor and non-inhibitor molecules from the test set of the CYP3A4 dataset for KPGT and baseline methods, using t-distributed stochastic neighbor embedding (t-SNE).

Fig. R6: Prediction performance of KPGT and baseline methods evaluated exclusively on the activity cliffs in the test set of the CYP3A4 dataset, measured in terms of AUPRC. The results were reported based on three independent runs.

ated as Contextpred_{Sup}). We acknowledge that we did not explicitly define these abbreviations in our original manuscript, which led to the confusion. However, we want to clarify that the method Supervised+ContextPred mentioned by the reviewer was indeed included as a baseline in our comparison tests, as indicated by the red dashed boxes in Fig. R7. To enhance clarity and address this issue, we have revised Supplementary Note 1.2 to provide explicit definitions of the abbreviations of these baseline models, including Supervised+Context Prediction. We believe

that this clarification will alleviate any confusion regarding baseline methods. The relevant texts in Supplementary Note 1.2 (Lines 92-103, Page 3) are presented below for review:

“Attribute Masking (Masking) pre-trains a GIN through learning to predict the masked features of nodes²⁸. Context Prediction (Contextpred) pre-trains a GIN through learning to predict the surrounding graph structures of subgraphs²⁸. Supervised+Masking (Masking_{Sup}) pre-trains a GIN through the Masking strategy following a supervised graph-level bio-activities prediction task²⁸. Supervised+Context Prediction (Contextpred_{Sup}) pre-trains a GIN using the Contextpred strategy and subsequently pre-trains with a supervised graph-level bio-activities prediction task²⁸.”

Fig. R7: Benchmarking the performance of KPGT against nineteen self-supervised learning baseline methods. **a-b** The averaged prediction performance of KPGT and the self-supervised learning baseline methods for classification (measured by AUROC) and regression datasets (measured by RMSE) in two settings: (a) feature extraction, where the pre-trained LiGhT model remains fixed, and (b) transfer learning, where the pre-trained LiGhT model is trainable. These results were reported based on three independent runs with different random seeds.

5. I am curious to know whether cross-validation was used in the results presented in Figures 2(a) and 2(b). If cross-validation was employed, it is necessary to provide details about the specific cross-validation procedure utilized. On the other hand, if cross-validation was not utilized, it is recommended to incorporate cross-validation into the analysis.

Response: We apologize for the lack of details regarding the experimental settings of the first benchmarking test in the Results section. In our first benchmarking tests in the Results section, we conducted tests on each dataset three times, utilizing different random seeds for both data splitting and model training, and subsequently reported the averaged performance derived from these three separate splits. Following established practice from previous research^{14,34,35},

a scaffold splitting scheme was employed in this process to partition each dataset into training, validation, and test sets within an 8:1:1 ratio. This splitting scheme ensures the molecules in the test sets differ structurally from those in the training set, offering an ideal scenario to evaluate the robustness and generalizability of prediction models. In the revised manuscript, we have incorporated the necessary clarifications to illuminate the experimental settings. Furthermore, to enhance the organization of our manuscript, we have relocated the content pertaining to experimental settings to the subsection titled “Experimental settings of downstream molecular property prediction tasks” within the Methods section (Lines 460-498, Page 17). Notably, we have also introduced error bars to the bar plots to visually convey the degree of deviation observed in the results, as shown in Fig. R1.

Moreover, it is important to clarify that we did not employ conventional k-fold cross-validation in our tests. In k-fold cross-validation, the data is first randomly shuffled and divided into k equally-sized folds. The model is then trained and evaluated k times, with each fold serving as the test set while the remaining k-1 folds act as the training set. However, regarding molecular property prediction tasks, it has been extensively discussed in the literature^{14,34,35} that random splitting, as performed in the k-fold cross-validation, may not provide an ideal evaluating setting. In real drug discovery scenarios, the test molecules may possess distinct structural characteristics compared to the molecules in the training set. To properly assess the practical applicability of our proposed method, we adhered to scaffold splitting, a technique recommended in previous research^{14,34,35}. This approach splits molecules based on their structural scaffolds, ensuring that the molecules present in the test set differ structurally from those within the training set. Through the application of this data splitting approach, we aim to reasonably evaluate the capability of a model to generalize effectively to out-of-distribution samples, which is crucial for assessing its practical utility.

6. Figure 4 illustrates the identification of HPK1 inhibitors by KPGT. To enhance the presentation, it is suggested to include the docking score of the molecule identified by KPGT.

Response: We thank the reviewer for the constructive suggestion. The execution of docking analyses on the molecules identified by KPGT stands as a crucial step to further substantiate their potential for inhibiting HPK1. Following the reviewer’s suggestion, we have further carried out docking analyses on the molecules identified by KPGT that potentially inhibit HPK1. Autodock Vina^{36,37}, a widely used docking software, was employed for these tests. The reference protein-ligand structure (PDB ID: 7SIU³⁸) was used to guide the identification of the binding pocket. As depicted in Fig. R8a, all the molecules achieved docking energies below -7 kcal/mol, a commonly used threshold for drug-like molecules³⁹⁻⁴², signifying substantial potential for these molecules as HPK1 inhibitors. Additionally, we conducted an in-depth analysis of the protein-ligand interactions for all the molecules that had not been reported in the literature, using a widely used protein-ligand interaction profiler called PLIP⁴³. Fig. R8b illustrates the protein-ligand interaction profile of the ligand gilteritinib with the protein HPK1. The analysis revealed the formation of three hydrophobic interactions and six hydrogen bonds between the ligand and the protein. Remarkably, the hydrogen bonds formed with residues 94A and 97A were also reported in the reference protein-ligand structure (PDB ID: 7SIU³⁸). These observations showcased that the molecules can tightly bind to HPK1, validating the reliability of the docking results. Fig. R10 provides additional protein-ligand interaction profiles for other molecules, including palbociclib, ripretinib, trilaciclib, rucaparib, selpercatinib, alatrofloxacin, and vericiguat. Collectively, these results highlighted the superior ability of KPGT to identify potential inhibitors for HPK1.

Additionally, we conducted docking analyses on the molecules identified by KPGT that potentially inhibit FGFR1. The protein-ligand structure (PDB ID: 5A4C⁴⁴) was employed as a reference for the binding pocket identification. As shown in Fig. R9a, all the top twenty molecules identified by KPGT achieve docking energies below -7 kcal/mol. By profiling the protein-ligand interactions utilizing PLIP⁴³, the ligand brigatinib tightly bound to the protein FGR1. Specifically, it formed four hydrophobic interactions, one hydrogen bond, and one salt bridge with FGFR1 (Fig. R9b). Among these interactions, the hydrogen bond formed with residue 641A was also reported in the reference protein-ligand structure (PDB ID: 5A4C⁴⁴). Additionally, in Fig. R11, we display additional protein-ligand interaction profiles for other molecules, including ripretinib, encorafenib, elagolix, baricitinib, enasidenib, and ruxolitinib. All these observations collectively reinforced the generalizability of KPGT in accelerating the identification of potential drug candidates, thus establishing its utility as a valuable tool in drug discovery.

In our revised manuscript, we have integrated the results of docking analyses into Figs. 4-5 and Supplementary Figs. 16-17, and have also made corresponding updates to the relevant texts in the subsection named "Uncovering potentially effective inhibitors for antitumor targets by KPGT" within the Results section (Lines 271-285, Page 11 and Lines 296-304, Page 12).

Fig. R8: Docking analyses of the potential inhibitors against HPK1 identified by KPGT. **a** The docking scores of the top twenty potential inhibitors identified by KPGT, measured by Autodock Vina^{36,37}. **b** The interactions between giliteritnib and HPK1 profiled by PLIP⁴³. The protein-ligand structure (PDB ID: 7SIU³⁸) was utilized as a reference for the identification of the binding pocket. The red line at -7 kcal/mol represents a commonly used threshold for identifying drug-like molecules.

7. Figure 2(d) displays the RMSE score of $MoleculeACE_{All}$ and $MoleculeACE_{Cliff}$ under KPGT, SVM, GBM and RF. It is worth noting that while SVM, GBM and RF are machine learning methods, KPGT is a deep neural network. To ensure a comprehensive comparison, the experiments involving $MoleculeACE_{All}$ and $MoleculeACE_{Cliff}$ should encompass the deep learning method, and the machine learning method should be added to the Supplementary file.

Response: We appreciate the valuable suggestion provided by the reviewer. Regarding the results on the datasets from the MoleculeACE benchmarking platform²⁶, we aimed to enhance clarity by showcasing the results of the most competitive baseline methods, that is, support vector machine (SVM)⁴⁵, gradient boosting machines (GBM)⁴⁶, and random forest (RF)⁴⁷, in Fig. 2d-e within our original manuscript. However, it is crucial to note that our comparison of KPGT

Fig. R9: Docking analyses of the potential inhibitors against FGFR1 identified by KPGT. **a** The docking scores of the top twenty potential inhibitors identified by KPGT, measured by Autodock Vina^{36,37}. **b** The interactions between brigatinib and FGFR1 profiled by PLIP⁴³. The protein-ligand structure (PDB ID: 5A4C⁴⁴) was utilized as a reference for the identification of the binding pocket. The red line at -7 kcal/mol on the y-axis represents a commonly used threshold for identifying drug-like molecules.

extended to all twenty-four baseline methods introduced in the MoleculeACE benchmarking platform, including sixteen machine learning-based methods and eight deep learning-based methods. The complete results of all baseline methods can be found in Supplementary Figs. 4-5 within our original supplementary information. In our revised manuscript, we have reported the averaged prediction performance of KPGT and all baseline methods across all datasets in Fig. 2d-e. Additionally, to enable a more comprehensive evaluation of KPGT on the datasets from MoleculeACE, we have included the nineteen self-supervised learning-based methods employed in the first benchmarking test in the Results section. Fig. R2a and Table R2 report the results of KPGT and baseline methods under the MoleculeACE_{All} setting. The results demonstrated that KPGT surpassed baseline methods on twenty-six out of thirty datasets, with an overall relative improvement of 3.9%. Fig. R2b and Table R3 report the results of KPGT and baseline methods evaluated under the MoleculeACE_{Cliff} setting (exclusive evaluation on activity cliffs within the test set). From the results, KPGT achieved superior prediction performance in comparison with baseline methods on twenty-two out of thirty datasets, exhibiting an overall relative improvement of 1.2%. These observations demonstrated the efficacy of KPGT in predicting molecular bio-activities, even when handling activity cliffs. We have made revisions to the Results section of our manuscript to incorporate these new results (Lines 161-165, Page 7) and updated these results to Fig. 2.

- To the best of my understanding, the knowledge utilized in this paper refers to molecular descriptors and fingerprints. However, there seems to be an issue regarding the definitions of these molecular descriptors and fingerprints. According to lines 156-157, the definitions should be presented in Notes 1.2. However, upon reviewing S1.2 in the Supplementary file, it appears that it provides only a brief introduction to the baseline methods instead. It is crucial to accurately describe the definitions of molecular descriptors and fingerprints in the appropriate section.

Response: We apologize for the lack of clarity regarding the definitions of “knowledge”, “molecular descriptors”, and “fingerprints” in the initial version of our manuscript. These fundamental aspects of our study require explicit definitions to foster a better comprehension of our work. To

Table R2: Prediction performance of KPGT and the best baseline method on the datasets from MoleculeACE under the MoleculeACE_{All} setting, measured in terms of RMSE. The best result for each dataset is marked in bold.

Dataset	Current best baseline		KPGT	
	Method	Score	Score	Rank
CHEMBL1862_Ki	SVM _{ECFP}	0.668	0.633	1st
CHEMBL1871_Ki	SVM _{ECFP}	0.668	0.605	1st
CHEMBL2034_Ki	SVM _{ECFP}	0.696	0.679	1st
CHEMBL204_Ki	SVM _{ECFP}	0.614	0.588	1st
CHEMBL2047_EC50	SVM _{ECFP}	0.705	0.666	1st
CHEMBL214_Ki	SVM _{ECFP}	0.612	0.587	1st
CHEMBL2147_Ki	SVM _{ECFP}	0.683	0.652	1st
CHEMBL218_EC50	RF _{MACCS}	0.666	0.625	1st
CHEMBL219_Ki	SVM _{ECFP}	0.743	0.718	1st
CHEMBL228_Ki	SVM _{ECFP}	0.704	0.669	1st
CHEMBL231_Ki	RF _{MACCS}	0.705	0.610	1st
CHEMBL233_Ki	SVM _{ECFP}	0.740	0.691	1st
CHEMBL234_Ki	SVM _{ECFP}	0.637	0.606	1st
CHEMBL235_EC50	RF _{ECFP}	0.637	0.624	1st
CHEMBL236_Ki	SVM _{ECFP}	0.692	0.655	1st
CHEMBL237_EC50	SVM _{ECFP}	0.760	0.716	1st
CHEMBL237_Ki	SVM _{ECFP}	0.707	0.678	1st
CHEMBL238_Ki	GBM _{ECFP}	0.611	0.537	1st
CHEMBL239_EC50	SVM _{ECFP}	0.681	0.644	1st
CHEMBL244_Ki	SVM _{ECFP}	0.738	0.698	1st
CHEMBL262_Ki	RF _{ECFP}	0.703	0.627	1st
CHEMBL264_Ki	SVM _{ECFP}	0.583	0.574	1st
CHEMBL2835_Ki	RF _{ECFP}	0.410	0.373	1st
CHEMBL287_Ki	GBM _{MACCS}	0.767	0.706	1st
CHEMBL2971_Ki	GBM _{ECFP}	0.606	0.571	1st
CHEMBL3979_EC50	GBM _{ECFP}	0.686	0.669	1st
CHEMBL4005_Ki	SVM _{ECFP}	0.550	0.559	2nd
CHEMBL4203_Ki	SVM _{ECFP}	0.880	0.830	1st
CHEMBL4616_EC50	SVM _{ECFP}	0.589	0.587	1st
CHEMBL4792_Ki	SVM _{ECFP}	0.675	0.619	1st

Table R3: Prediction performance of KPGT and the best baseline methods on the datasets from the MoleculeACE benchmark under the MoleculeACE_{Cliff} setting, measured in terms of RMSE. The best result for each dataset is marked in bold.

Dataset	Current best baseline		KPGT	
	Method	Score	Score	Rank
CHEMBL1862_Ki	SVM _{ECFP}	0.634	0.633	1st
CHEMBL1871_Ki	RF _{ECFP}	0.709	0.701	1st
CHEMBL2034_Ki	SVM _{ECFP}	0.842	0.806	1st
CHEMBL204_Ki	SVM _{PHYSCHEM}	0.667	0.654	1st
CHEMBL2047_EC50	SVM _{ECFP}	0.777	0.747	1st
CHEMBL214_Ki	SVM _{ECFP}	0.7	0.694	1st
CHEMBL2147_Ki	SVM _{ECFP}	0.736	0.705	1st
CHEMBL218_EC50	RF _{MACCS}	0.733	0.701	1st
CHEMBL219_Ki	SVM _{ECFP}	0.779	0.727	1st
CHEMBL228_Ki	GBM _{ECFP}	0.755	0.759	2nd
CHEMBL231_Ki	GBM _{PHYSCHEM}	0.772	0.774	2nd
CHEMBL233_Ki	SVM _{ECFP}	0.787	0.731	1st
CHEMBL234_Ki	SVM _{ECFP}	0.683	0.646	1st
CHEMBL235_EC50	RF _{ECFP}	0.761	0.759	1st
CHEMBL236_Ki	SVM _{ECFP}	0.836	0.793	1st
CHEMBL237_EC50	SVM _{MACCS}	0.842	0.785	1st
CHEMBL237_Ki	SVM _{ECFP}	0.754	0.706	1st
CHEMBL238_Ki	GBM _{ECFP}	0.696	0.657	1st
CHEMBL239_EC50	SVM _{ECFP}	0.731	0.746	2nd
CHEMBL244_Ki	SVM _{ECFP}	0.835	0.82	1st
CHEMBL262_Ki	SVM _{ECFP}	0.655	0.601	1st
CHEMBL264_Ki	SVM _{ECFP}	0.668	0.642	1st
CHEMBL2835_Ki	SVM _{MACCS}	0.716	0.666	1st
CHEMBL287_Ki	SVM _{ECFP}	0.772	0.729	1st
CHEMBL2971_Ki	GBM _{ECFP}	0.739	0.792	4th
CHEMBL3979_EC50	SVM _{ECFP}	0.727	0.723	1st
CHEMBL4005_Ki	SVM _{ECFP}	0.647	0.673	2nd
CHEMBL4203_Ki	SVM _{ECFP}	1.001	1.114	7th
CHEMBL4616_EC50	SVM _{ECFP}	0.618	0.583	1st
CHEMBL4792_Ki	SVM _{ECFP}	0.741	0.709	1st

Fig. R10: Docking results of the previously unreported potential HPK1 inhibitory molecules identified through KPGT. The interactions between molecules and HPK1 were profiled by PLIP⁴³. The protein-ligand structure (PDB ID: 7SIU³⁸) was utilized as a reference for the identification of the binding pocket.

address this concern, we have made revisions to our manuscript, specifically within the Introduction section, to provide the definitions for these terms. In our study, the term “knowledge” refers to any quantifiable characteristics of molecules (also elaborated in our response to Point 9). Meanwhile, “molecular descriptors” are quantitative descriptions of the physical and chemical profiles of molecules, generated by logical and mathematical procedures⁴⁸. For example, LogP is a type of molecular descriptor measuring the lipophilicity of molecules. On the other hand, molecular fingerprints are bit strings that characterize molecular structures, where one indicates the existence of one substructure and zero otherwise^{32,49}. As presented in our manuscript, our proposed method employs 200 molecular descriptors and 512 RDKit fingerprints. Table R4 offers a comprehensive listing of these molecular descriptors, and Fig. R12 presents examples of RDKit fingerprints. In our revised manuscript, we have incorporated these results into Supplementary Table 14 and Supplementary Fig. 21.

Furthermore, to assist users in generating these molecular descriptors and fingerprints, we have provided the scripts and detailed instructions in our released code, which can be accessed at <https://github.com/lihan97/KPGT>.

Fig. R11: Docking results of previously unreported potential FGFR1 inhibitory molecules identified through KPGT. The interactions between molecules and FGFR1 were profiled by PLIP⁴³. The protein-ligand structure (PDB ID: 5A4C³⁸) was utilized as a reference for the identification of the binding pocket.

9. Lines 186-188 indicate that additional knowledge is utilized in the self-supervised learning methods Contextpredsup, Maskingsup, and GROVER. However, it is unclear what specific types of knowledge are employed in these methods. Are you referring to context, masked nodes or edges, or other forms of knowledge? It is important to note that the author has previously defined molecule descriptors and fingerprints as knowledge. Masked nodes, edges, context, and properties cannot be classified as knowledge in the same way. If you wish to include these elements as part of the knowledge in this paper, it is necessary to clearly define the scope of knowledge within the context of this study.

Response: We apologize for any confusion caused by the lack of clarity in the definition of “knowledge” in the initial version of our manuscript. As previously outlined in our response to Point 8, we have addressed this issue by presenting a clear definition of knowledge within the revised Introduction section. In the context of our study, the term “knowledge” refers to any quantifiable characteristics of molecules. This encompasses various facets, such as the aforementioned molecular descriptors and fingerprints, which offer valuable insights into the features of molecules and

Table R4: The 200 molecular descriptors employed in KPGT.

BalabanJ	BertzCT	Chi0	Chi0n	Chi0v
Chi1	Chi1n	Chi1v	Chi2n	Chi2v
Chi3n	Chi3v	Chi4n	Chi4v	EState_VSA1
EState_VSA10	EState_VSA11	EState_VSA2	EState_VSA3	EState_VSA4
EState_VSA5	EState_VSA6	EState_VSA7	EState_VSA8	EState_VSA9
ExactMolWt	FpDensityMorgan1	FpDensityMorgan2	FpDensityMorgan3	FractionCSP3
HallKierAlpha	HeavyAtomCount	HeavyAtomMolWt	lpc	Kappa1
Kappa2	Kappa3	LabuteASA	MaxAbsEStateIndex	MaxAbsPartialCharge
MaxEStateIndex	MaxPartialCharge	MinAbsEStateIndex	MinAbsPartialCharge	MinEStateIndex
MinPartialCharge	MolLogP	MolMR	MolWt	NHOHCount
NOCCount	NumAliphaticCarbocycles	NumAliphaticHeterocycles	NumAliphaticRings	NumAromaticCarbocycles
NumAromaticHeterocycles	NumAromaticRings	NumHAcceptors	NumHDonors	NumHeteroatoms
NumRadicalElectrons	NumRotatableBonds	NumSaturatedCarbocycles	NumSaturatedHeterocycles	NumSaturatedRings
NumValenceElectrons	PEOE_VSA1	PEOE_VSA10	PEOE_VSA11	PEOE_VSA12
PEOE_VSA13	PEOE_VSA14	PEOE_VSA2	PEOE_VSA3	PEOE_VSA4
PEOE_VSA5	PEOE_VSA6	PEOE_VSA7	PEOE_VSA8	PEOE_VSA9
RingCount	SMR_VSA1	SMR_VSA10	SMR_VSA2	SMR_VSA3
SMR_VSA4	SMR_VSA5	SMR_VSA6	SMR_VSA7	SMR_VSA8
SMR_VSA9	SlogP_VSA1	SlogP_VSA10	SlogP_VSA11	SlogP_VSA12
SlogP_VSA2	SlogP_VSA3	SlogP_VSA4	SlogP_VSA5	SlogP_VSA6
SlogP_VSA7	SlogP_VSA8	SlogP_VSA9	TPSA	VSA_EState1
VSA_EState10	VSA_EState2	VSA_EState3	VSA_EState4	VSA_EState5
VSA_EState6	VSA_EState7	VSA_EState8	VSA_EState9	fr_Al_COO
fr_Al_OH	fr_Al_OH_noTert	fr_ArN	fr_Ar_COO	fr_Ar_N
fr_Ar_NH	fr_Ar_OH	fr_COO	fr_COO2	fr_C_O
fr_C_O_noCOO	fr_C_S	fr_HOCCN	fr_Imine	fr_NH0
fr_NH1	fr_NH2	fr_N_O	fr_Ndealkylation1	fr_Ndealkylation2
fr_Nhpyrrole	fr_SH	fr_aldehyde	fr_alkyl_carbamate	fr_alkyl_halide
fr_allylic_oxid	fr_amide	fr_amidine	fr_aniline	fr_aryl_methyl
fr_azide	fr_azo	fr_barbitur	fr_benzene	fr_benzodiazepine
fr_bicyclic	fr_diazo	fr_dihydropyridine	fr_epoxide	fr_ester
fr_ether	fr_furan	fr_guanido	fr_halogen	fr_hdrzine
fr_hdrzone	fr_imidazole	fr_imide	fr_isocyan	fr_isothiocyan
fr_ketone	fr_ketone_Topless	fr_Lactam	fr_Lactone	fr_methoxy
fr_morpholine	fr_nitrile	fr_nitro	fr_nitro_ arom	fr_nitro_ arom_nonortho
fr_nitroso	fr_oxazole	fr_oxime	fr_para_hydroxylation	fr_phenol
fr_phenol_noOrthoHbond	fr_phos_acid	fr_phos_ester	fr_piperdine	fr_piperzine
fr_priamide	fr_prisulfonamd	fr_pyridine	fr_quatN	fr_sulfide
fr_sulfonamd	fr_sulfone	fr_term_acetylene	fr_tetrazole	fr_thiazole
fr_thiocyan	fr_thiophene	fr_unbrch_alkane	fr_urea	qed

serve as prevalent tools for representing their inherent characteristics. Additionally, knowledge also includes experimentally derived characteristics of molecules. For instance, the preprocessed ChEMBL dataset⁵⁰ provides measurements of bioactivities for 456,000 molecules across 1,310 bioassays. In our proposed method, we specifically considered molecular descriptors and fingerprints as additional knowledge. The reason behind this choice is that most molecular descriptors and fingerprints are readily available through existing cheminformatics tools^{33,51,52}. Hence, incorporating them into the pre-training process does not require any extra resources or budget. On the other hand, obtaining experimentally measured characteristics for a large number of molecules (e.g., the two million molecules used in our pre-training process) is not practical.

As discussed in our manuscript, the baseline methods Contextpred_{Sup}¹⁴, Masking_{Sup}¹⁴, and GROVER¹⁹ also utilize additional knowledge to enhance their prediction performance on molecu-

Fig. R12: An example of RDKit fingerprints of epinephrine. The numbers below the substructures indicate the positions of the corresponding fingerprints in the corresponding bit string.

lar properties. Contextpred_{Sup} and Masking_{Sup} ¹⁴ follow a two-stage pre-training process. The initial stage employs self-supervised pre-training schemes, namely context prediction and attribute masking. The subsequent stage integrates the preprocessed ChEMBL dataset⁵⁰ mentioned earlier and performs supervised pre-training by predicting molecular bio-activities. GROVER¹⁹ leverages additional knowledge during both the pre-training and fine-tuning processes. The pre-training phase of GROVER involves learning to predict the presence of molecular motifs. Then, during fine-tuning, GROVER integrates 200 molecular descriptors by concatenating them with the molecular representations provided by the pre-trained model to generate the final predictions. Therefore, Contextpred_{Sup} and Masking_{Sup} incorporate bio-activity measurements from ChEMBL dataset as additional knowledge, while GROVER utilizes molecular motifs and molecular descriptors as additional knowledge. To ensure accuracy in the description of incorporating additional knowledge in the baseline methods, we have carefully revised Supplementary Note 1.2.

- Overall, this paper is readable; however, there are issues with the logic and framework as they are not well structured.

Response: We express our sincere appreciation to the reviewer for providing valuable feedback concerning the readability of our manuscript. In our continuous effort to enhance the logical coherence and structural organization of our manuscript, we have undergone an extensive rewriting process to deliver a clearer and more cohesive presentation. Beginning with the Introduction section, we conducted a comprehensive revision that entails a detailed exploration of the evolution of AI-based approaches for molecular property prediction. This discussion now serves as a potent motivator for the introduction of our proposed method. Furthermore, we have introduced explicit definitions of the terms "knowledge", "molecular descriptors", and "fingerprint" within this section, to facilitate a clearer grasp of these crucial terms that are employed throughout our manuscript. In the Results section, particularly within the subsection titled "Overview of KPGT", we have properly simplified the language while ensuring the preservation of the core contributions of our proposed method. Furthermore, for other subsections within the Results section, we have provided detailed descriptions of experimental settings, allowing readers to more comprehensively grasp

our experimental setup. Notably, the content related to these experimental settings has been relocated to the subsection titled “Experimental settings of downstream molecular property prediction tasks” within the Methods section, optimizing the conciseness and clarity of these subsections. Moreover, we have enhanced the presentation of our experimental results. This refinement ensures that significant findings receive detailed elaboration, effectively highlighting the primary advancement of our proposed method. Additionally, we have conducted a range of supplementary computational tests, encompassing evaluation tests, visualizations, docking analyses, and ablation studies within the Results section, to ensure the robustness and rigor of our conclusions. Concurrently, we have enhanced the transitions between different experimental segments, thereby ensuring a seamless progression of ideas.

Moreover, we have included a Discussion section to summarize the primary contributions of our proposed method. Simultaneously, we have discussed the potential limitations inherent in our approach, thereby illuminating the future research directions. With these refinements, we firmly believe that our rigorous revision process has resulted in a logical flow that is more coherent and easily comprehensible. We provide the newly added Discussion section below for review:

“In this study, we develop and establish KPGT, a self-supervised learning framework that offers improved, generalizable, and robust molecular property prediction through significantly enhanced molecular representation learning. By leveraging a high-capacity backbone model called LiGhT, KPGT comprehensively captures the inherent structural information within molecular graphs. More importantly, KPGT introduces a knowledge-guided pre-training strategy that can robustly address the limitations of previously ill-defined pre-training approaches, empowering our model to provide semantic-enriched molecular representations. In addition, KPGT incorporates several finetuning strategies that effectively integrate the acquired knowledge from the pre-trained model, leading to improved performance on downstream molecular property prediction tasks. With these advancements, KPGT achieved great improvement on sixty-three datasets compared with several baseline methods. Remarkably, KPGT showed practical applications in the identification of multiple potential inhibitors for two antitumor targets.

Despite the advantages of KPGT for effective molecular property prediction, there remain a few limitations. First, the integration of additional knowledge is the most distinguishable feature of our proposed method. Except for the 200 molecular descriptors and 512 RDKFP employed in KPGT, there is potential to incorporate various other types of additional informative knowledge. For instance, Mordred¹³ can calculate over 1,800 molecular descriptors, presenting an opportunity to incorporate a broader range of knowledge to further enrich the representation learning of molecules. Moreover, further studies could encompass the integration of three-dimensional (3D) molecular conformations into the pre-training process, thus enabling the model to capture vital 3D information regarding molecules, and potentially enhancing the representation learning capabilities. Additionally, while KPGT currently employs a backbone model with approximately one hundred million parameters, along with pre-training on two million molecules, exploring larger-scale pre-training could offer even more substantial benefits for molecular representation learning. Overall, we anticipate our proposed method will offer a general self-supervised learning framework for accelerating AI-aided drug discovery.”

11. There are also some grammar errors in this article.

Response: We express our gratitude to the reviewer for providing such valuable feedback. To address this concern, we have undertaken a comprehensive review of our manuscript and con-

ducted meticulous proofreading across its entirety, ensuring the accuracy and precision of the language used.

Reviewer 2

1. Remarks to the Author: This paper proposed a novel knowledge-guided pre-training of graph transformer, named (KPGT), for learning molecular representations. KPGT combines a graph transformer designed for molecular graphs with a knowledge-guided pre-training strategy to capture molecule knowledge. The authors evaluate their method on various datasets and compare it with several baseline methods. The results demonstrated the superior performance of KPGT on predicting molecular properties. Overall, the paper is interesting and well-presented. However, the novelty appears a bit weak as some self-supervised pre-training frameworks similar to KPGT have been previously developed for molecule representation learning, such as Fang et al., 2022, Xu et al., 2021, and Rong et al, 2020 (see below for the reference). There are a few concerns that should be addressed to strengthen the study.

Response: We sincerely appreciate the reviewer's insightful summary of our work. It is gratifying to know that the reviewer found our paper interesting and well-presented. Moreover, we acknowledge the reviewer's concern regarding the novelty of our proposed method within the domain of self-supervised learning frameworks for molecule representation learning. As the reviewer pointed out, there indeed exist several self-supervised learning-based approaches, similar to KPGT, that have been proposed for molecular property prediction^{14–23}. As discussed in the Introduction section of our manuscript, these methods typically involve the modification of molecular graphs by means of node or subgraph masking, followed by the prediction of the masked components, or the utilization of contrastive learning objectives to align the modified graphs with their corresponding originals in the latent space. It is worth noting that the methods mentioned by the reviewer, namely GEM²³, GraphLoG¹⁷, and GROVER¹⁹, also utilize similar strategies in their frameworks. Molecules inherently possess characteristics tightly linked to their structures, which implies that even minor modifications of molecular graphs can lead to the loss of their semantic information. Therefore, existing methods primarily model the structural similarity of molecules but lack the capability to capture the relationships between molecular structures and their semantics, which plays a crucial role in downstream molecular property prediction tasks. To address this issue, we introduced additional knowledge to initialize a knowledge node, which serves as an effective source of semantic information, guiding the model in capturing the intricate relationships between molecular structure and semantics. To the best of our knowledge, KPGT is the *first* framework to employ additional knowledge to guide the self-supervised learning on molecules (as elaborated in our response to Point 9). Additionally, KPGT incorporates a high-capacity backbone network, named LiGhT, which is also the *first* graph transformer to model molecular line graphs, fully capturing the structural information of molecular graphs. Furthermore, in our computational tests, KPGT consistently achieved superior prediction performance compared to self-supervised learning-based baseline methods (including the methods mentioned by the reviewer, namely GEM²³, GraphLoG¹⁷, and GROVER¹⁹) on molecular property prediction tasks across various domains (as elaborated in our response to Point 4). Our ablation studies also highlighted the significant contributions of the individual design components of KPGT (as elaborated in our response to Point 7). These findings collectively provided sufficient evidence supporting the significant contribution of KPGT in modeling molecular properties, not only in terms of the novelty of the proposed self-supervised learning framework but also in its superior prediction performance. We have carefully considered the reviewer's comments and taken them into account to strengthen the quality and impact of our study. Below, we will provide a detailed response addressing each

of the reviewer's comments.

2. Clarify Splitting Schemes: The authors use different splitting schemes for different tasks, which may lead to confusion. For example, the authors divide the dataset into training, validation, and test sets with an 8:1:1 ratio in the classification task, while it is divided with a 7:1:2 ratio in the molecular property prediction tasks. It would be helpful to provide a clear explanation of the rationale behind these choices and clarify whether the proposed model is robust against different splitting schemes.

Response: We apologize for any confusion caused by the different splitting schemes used for datasets from different benchmarks. First, we would like to explain the rationale behind these choices. We employed molecular property datasets from three benchmarks to comprehensively evaluate the prediction performance of our proposed method. For the first benchmarking test in the Results section, we employed a scaffold splitting scheme with an 8:1:1 ratio (train:validation:test) following the established practices in previous research^{14,34,35}. This splitting scheme divides the dataset based on molecular scaffolds, ensuring that similar scaffolds are not present in both training and test sets. This approach enables one to assess the generalization ability of a model to out-of-distribution samples, which is crucial for evaluating its practical utility. In the case of the dataset from the TDC and MoleculeACE benchmarking platforms, we rigorously adhered to the evaluation protocols established by the respective platforms, which was essential to guarantee equitable comparisons between KPGT and the baseline methods offered by these platforms. Specifically, for the datasets from the TDC benchmarking platform, we employed a scaffold splitting scheme with a 7:1:2 ratio (train:validation:test) in accordance with the evaluation protocol of TDC. As for the datasets from the MoleculeACE benchmarking platform, we employed a stratified random splitting scheme with the same 7:2 ratio (train:test), following the evaluation protocol of MoleculeACE. Thus, different splitting schemes were used for these different benchmarking tests. Next, we would like to clarify the robustness of our proposed model to different splitting schemes. As mentioned earlier, a scaffold splitting scheme was employed for the datasets from the first benchmarking tests and the TDC benchmarking platform, while a stratified random splitting scheme was employed for the datasets from the MoleculeACE benchmarking platform. Overall, KPGT achieved superior prediction performance under these different splitting schemes (Fig. 2). Furthermore, in the subsection titled "Uncovering potentially effective inhibitors for antitumor targets by KPGT" within the Results section, we evaluated the performance of KPGT in predicting inhibitors for antitumor targets across three distinct scenarios: scaffold splitting, time splitting, and domain transfer (Figs. 4-5). In each of these scenarios, KPGT consistently outperformed the baseline methods. Therefore, we can conclude that KPGT exhibits robustness against distinct splitting schemes and different application scenarios.

3. Include Baseline Results: The results of many baselines are missing from the manuscript. For example, in the Results section, the authors claim that they compare KPGT with 17 baseline methods on 22 molecular property prediction tasks from the TDC benchmark, while with 24 baseline methods on the datasets from the MoleculeACE benchmark. Unfortunately, the authors only show the results of the best baseline methods evaluated by the benchmarking papers. It would be beneficial to include the results of all evaluated baseline methods for a more comprehensive comparison.

Response: We appreciate the valuable suggestion provided by the reviewer. In response to this suggestion, we have made necessary revisions to our manuscript. In particular, we have included the results of all evaluated baseline methods on the datasets from the TDC^{53,54} and MoleculeACE²⁶ benchmarking platforms in our revised manuscript. These results can be found in Supplementary Figs. 4-8 and Figs. 2d-e. We have also attached these figures below for review (Fig. R13-R18). To facilitate a more comprehensive comparison, we have also provided a folder “Source Data” that contains the raw results of KPGT and baseline methods for all the benchmarking tests.

Fig. R13: Prediction performance of KPGT and baseline methods on the datasets measuring absorption from the TDC benchmarking platform.

4. Compare with Advanced Models: The authors claim that “These results suggested that the previously designed deep learning models might fail to capture the intrinsic features of molecules”. It seems that most of the baseline methods used in the benchmarking papers are traditional methods. Recently, more advanced self-supervised models for molecular property prediction have been proposed, such as Xu et al., 2021 and Fang et al. 2022. To support this claim and further demonstrate the superiority of KPGT, it would be valuable to compare it with more advanced self-supervised models.

Response: We thank the reviewer for the constructive suggestion. We concur with the reviewer’s perspective on the necessity of introducing more advanced self-supervised learning-based methods for a comprehensive evaluation of KPGT’s prediction performance. In the first benchmarking test in the Results section of our original manuscript, we conducted a comparative anal-

Fig. R14: Prediction performance of KPGT and baseline methods on the datasets measuring distribution from the TDC benchmarking platform.

Fig. R15: Prediction performance of KPGT and baseline methods on the datasets measuring metabolism from the TDC benchmarking platform.

ysis between KPGT and fourteen self-supervised learning-based baseline methods. Following the reviewer's suggestion, we have further enhanced our evaluation by including five additional advanced self-supervised learning-based methods: GEM²³, GraphMAE²⁴, MoleBERT²⁵, Supervised+Edgepred (Edgepred_{Sup})¹⁴, and Supervised+Infomax (Infomax_{Sup})¹⁴, to provide a more comprehensive assessment of KPGT. We would also like to clarify that GraphLoG¹⁷, which the reviewer recommended, had already been included as a baseline method in our original manuscript. In the feature extraction setting, KPGT exhibited superior performance compared to baseline methods across seven out of eight classification datasets and two out of three regression datasets, leading to overall relative improvements of 2.0% for classification and 4.5% for regression (Fig. R19a). In the finetuning setting, KPGT outperformed baseline methods across seven out of eight clas-

Fig. R16: Prediction performance of KPGT and baseline methods on the datasets measuring excretion from the TDC benchmarking platform.

Fig. R17: Prediction performance of KPGT and baseline methods on the datasets measuring toxicity from the TDC benchmarking platform.

sification datasets and all three regression datasets, resulting in overall relative improvements of 1.6% for classification and 4.2% for regression (Fig. R19b). These observations demonstrated the superior performance of KPGT in comparison to other self-supervised learning-based methods. In the revised manuscript, we have integrated the updated results into Fig. 2 and included the complete results in Supplementary Tables 4-7.

5. Provide Explanation for Figure 2c: "As shown in Figure 2c, KPGT achieved the best prediction performance on sixteen out of . . .". but it is unclear how this conclusion is made. Additional explanations regarding Figure 2c would help readers understand the authors' findings.

Response: We apologize for any confusion caused by Fig. 2c in our initial manuscript. Initially, we used a radar chart to visualize the ranking of KPGT on the performance leaderboards provided by the TDC benchmarking platform^{53,54}. To address the reviewer's concern, we have made revisions to enhance the readability and clarity of this figure. More specifically, we have replaced the original radar chart with a lollipop chart in the revised version of our manuscript, where the x-axis showcases different datasets and the y-axis presents the ranking of KPGT on the performance leaderboard of each dataset. We have incorporated the modifications into Fig. 2. The revised figure has been attached below for review (Fig. R21).

Fig. R18: Average prediction performance of KPGT and baseline methods on the datasets from the MoleculeACE benchmark under the (a) MoleculeACE_{All} and (b) MoleculeACE_{Cliff} settings, respectively, measured in terms of RMSE.

Fig. R19: Benchmarking the performance of KPGT against self-supervised learning baseline methods. **a-b** The averaged prediction performance of KPGT and self-supervised learning baseline methods on eight classification datasets and three regression datasets that had been widely used for evaluating self-supervised learning methods on molecules, measured in terms of AUROC and RMSE, respectively, under (a) the feature extraction setting (where the pre-trained LiGHT model is fixed) and (b) the transfer learning setting (where the pre-trained LiGHT model is trainable). The results were reported based on three independent runs with different random seeds.

Fig. R20: Averaged prediction performance of KPGT and baseline methods on the datasets from the MoleculeACE benchmarking platform under the (a) MoleculeACE_{All} and (b) $\text{MoleculeACE}_{Cliff}$ settings, respectively, measured in terms of RMSE.

Fig. R21: The ranking results of KPGT on the leaderboards from the TDC benchmarking platform for predicting absorption, distribution, metabolism, excretion, and toxicity properties of molecules. The ranking results were reported based on the averaged results derived from five independent runs. Dashed lines are used to separate different molecular property categories.

6. Justify Target Selection: In section 2.4, the authors take two antitumor targets HPK1 and FGFR1 as examples to evaluate the practical applicability of KPGT. It would be helpful to provide justification for choosing these targets and discuss whether the proposed model can be generalized to other targets.

Response: We appreciate the valuable suggestion provided by the reviewer. We are in full agreement with the reviewer's perspective that a robust justification is imperative for the selection of targets to be evaluated. In line with the reviewer's guidance, our revised manuscript now provides a more comprehensive rationale for the selection of these specific targets. Primarily, HPK1 and FGFR1 are widely recognized antitumor targets that have been extensively studied in the field^{55–58}. Their dysregulation has been implicated in various types of cancer, underscoring their

importance as therapeutic targets. Therefore, evaluating the performance of KPGT on these targets provides valuable insights into its potential applicability in facilitating the discovery of inhibitors against clinically relevant proteins. Second, the availability of high-quality experimental data for HPK1 and FGFR1 greatly facilitates the development and validation of our model. This thus provides adequate data for evaluating the practicality and prediction performance of KPGT. We have revised relevant paragraphs in the subsection titled “Uncovering potentially effective inhibitors for antitumor targets by KPGT” within the Results section (Lines 241-247, Page 9), which is presented as follows for review: “Hematopoietic progenitor kinase 1 (HPK1) and fibroblast growth factor receptor (FGFR1), which are implicated in a variety of cancer types, have been extensively studied for antitumor therapy⁷⁵⁻⁷⁸. The availability of high-quality experimental data for HPK1 and FGFR1 significantly facilitates the development and validation of the AI-based computation model, providing adequate data for evaluating the practicality and prediction performance of KPGT. In this section, we carried out evaluation tests, drug repurposing, and docking analyses for both targets, serving as a proof-of-concept validation of the effectiveness of KPGT in real-world drug discovery scenarios.”

Next, we would like to discuss the generalizability of KPGT to other targets. In addition to the antitumor targets discussed earlier, we engaged in a comprehensive evaluation of the prediction performance of KPGT for diverse targets encompassed within the datasets outlined in the Results section. To be more specific, the MoleculeACE benchmarking platform²⁶ provides a collection of datasets for thirty distinct protein targets, including the androgen receptor, ghrelin receptor, and janus kinase 1, among others. Based on the superior prediction performance exhibited by KPGT on these datasets, we believe that KPGT should be generalizable to other targets.

7. Conduct Ablation Study: In the proposed framework, the authors introduce path encoding and distance encoding modules, which, I think, are critical components of the proposed model. Conducting an ablation study to test the contributions of these modules to the model's performance would provide a deeper understanding of their effectiveness.

Response: We thank the reviewer for the constructive suggestion. We agree with the reviewer that conducting ablation studies is essential to comprehensively assess the contributions of our proposed path encoding and distance encoding modules. In Section 2.5 of our revised manuscript, we have conducted more comprehensive ablation studies on KPGT to validate the effectiveness of its specific design choices, including the path encoding and distance encoding modules, as suggested by the reviewer. Specifically, we have introduced several modified frameworks based on KPGT with specific restrictions: KPGT-KN (without knowledge nodes), KPGT-PE (without path encoding module), KPGT-DE (without distance encoding module), KPGT-LG (replacing molecular line graph with the molecular graph used in Graphormer²⁷), KPGT-LiGhT+Graphormer (replacing backbone model LiGhT with Graphormer), and KPGT-LiGhT+GIN (replacing backbone model LiGhT with GIN²⁸). Table R5 reports the performance of KPGT and modified frameworks on the datasets from our first benchmarking test in the Results section. Notably, KPGT surpassed KPGT-PE and KPGT-DE with overall relative improvements of 2.7% and 3.0%, respectively, providing empirical validation for the pivotal role played by the distance encoding and path encoding modules. Overall, KPGT outperformed all the modified frameworks, highlighting the significant contributions of individual design components of KPGT to its superior performance. These new results have been incorporated into Supplementary Table 13. The relevant paragraph in subsec-

tion named “Ablation studies on KPGT” within the Results section (Lines 307-325, Page 13) is provided below for review:

“To validate the effectiveness of the specific design choices of KPGT, we conducted comprehensive ablation studies. Specifically, we introduced several modified frameworks based on KPGT with specific restrictions: KPGT-Pretrain (without pre-training), KPGT-KN (without knowledge nodes), KPGT-PE (without path encoding module), KPGT-DE (without distance encoding module), KPGT-LG (replacing molecular line graph with original molecular graph), KPGT-LiGhT+Graphormer (replacing backbone model LiGhT with Graphormer⁵⁷), and KPGT-LiGhT+GIN (replacing backbone model LiGhT with GIN⁴⁵). Additionally, for a fair comparison, we ensured that all models had approximately the same number of parameters (around 3.5 million). Supplementary Table 13 reports the performance of KPGT and modified frameworks on the datasets from our first benchmarking test in the Results section. The results showcased the superiority of KPGT over KPGT-LG, resulting in an overall relative improvement of 2.3%. This observation demonstrated the enhanced informativeness of the molecular line graph utilized in KPGT, in contrast to the original molecular graph in Graphormer⁵⁷. KPGT also outperforms KPGT-LiGhT+Graphormer with an overall relative improvement of 1.8%, indicating the enhanced capacity of our proposed backbone model, LiGhT, in effectively capturing the inherent structural information in comparison with Graphormer. Moreover, KPGT surpassed KPGT-PE and KPGT-DE with overall relative improvements of 2.7% and 3.0%, respectively, providing empirical validation for the significant role played by the distance encoding and path encoding modules. Overall, KPGT outperformed all the modified frameworks, highlighting the significant contributions of the individual design components of KPGT to its superior performance.”

Table R5: Ablation studies of KPGT. Several modified frameworks based on KPGT with specific restrictions are introduced, including KPGT-Pretrain (without pre-training), KPGT-KN (without the knowledge nodes), KPGT-PE (without the path encoding module), KPGT-DE (without the distance encoding module), KPGT-LG (replacing molecular line graph with the molecular graph used in Graphormer), KPGT-LiGhT+Graphormer (replacing the backbone model LiGhT with Graphormer), and KPGT-LiGhT+GIN (replacing the backbone model LiGhT with GIN). The averaged results on the classification and regression datasets from the first benchmarking test in the Results section were reported, measured in terms of AUROC and RMSE, respectively. The numbers in brackets are the standard deviations across three independent runs. The best results are marked in bold.

Method	Classification dataset	Regression dataset
	AVG (AUROC)	AVG (RMSE)
KPGT-LiGhT+GIN	0.817 _(0.009)	1.229 _(0.257)
KPGT-LiGhT+Graphormer	0.822 _(0.007)	1.260 _(0.081)
KPGT-LG	0.824 _(0.006)	1.250 _(0.303)
KPGT-PE	0.822 _(0.007)	1.281 _(0.254)
KPGT-DE	0.823 _(0.012)	1.306 _(0.144)
KPGT-KN	0.819 _(0.008)	1.287 _(0.304)
KPGT-Pretrain	0.792 _(0.001)	1.469 _(0.302)
KPGT	0.829 _(0.011)	1.162 _(0.197)

8. Add a Discussion Section: It is recommended to add a dedicated 'Discussion' section to the manuscript. This section would allow the authors to discuss the advantages and limitations of the proposed method, as well as its potential applications in practical biological scenarios

Response: We appreciate the valuable suggestion provided by the reviewer. As suggested by the reviewer, the incorporation of a dedicated discussion section is indeed pivotal for summarizing the primary contributions and engaging in a comprehensive exploration of the potential limitations of our work. In light of this, we have included a Discussion section to summarize the primary contributions of our proposed method. Simultaneously, we discuss the potential limitations inherent in our approach, thereby illuminating the future research directions. The newly added Discussion section (Lines 335-359, Page 14) is provided below for review:

"In this study, we develop and establish KPGT, a self-supervised learning framework that offers improved, generalizable, and robust molecular property prediction through significantly enhanced molecular representation learning. By leveraging a high-capacity backbone model called LiGhT, KPGT comprehensively captures the inherent structural information within molecular graphs. More importantly, KPGT introduces a knowledge-guided pre-training strategy that can robustly address the limitations of previously ill-defined pre-training approaches, empowering our model to provide semantic-enriched molecular representations. In addition, KPGT incorporates several finetuning strategies that effectively integrate the acquired knowledge from the pre-trained model, leading to improved performance on downstream molecular property prediction tasks. With these advancements, KPGT achieved great improvement on sixty-three datasets compared with several baseline methods. Remarkably, KPGT showed practical applications in the identification of multiple potential inhibitors for two antitumor targets.

Despite the advantages of KPGT for effective molecular property prediction, there remain a few limitations. First, the integration of additional knowledge is the most distinguishable feature of our proposed method. Except for the 200 molecular descriptors and 512 RDChEMBL employed in KPGT, there is potential to incorporate various other types of additional informative knowledge. For instance, Mordred¹³ can calculate over 1,800 molecular descriptors, presenting an opportunity to incorporate a broader range of knowledge to further enrich the representation learning of molecules. Moreover, further studies could encompass the integration of three-dimensional (3D) molecular conformations into the pre-training process, thus enabling the model to capture vital 3D information regarding molecules, and potentially enhancing the representation learning capabilities. Additionally, while KPGT currently employs a backbone model with approximately one hundred million parameters, along with pre-training on two million molecules, exploring larger-scale pre-training could offer even more substantial benefits for molecular representation learning. Overall, we anticipate our proposed method will offer a general self-supervised learning framework for accelerating AI-aided drug discovery."

We believe that the inclusion of this Discussion section will comprehensively address the reviewer's suggestion and enhance the overall clarity and depth of our manuscript.

9. Justify "Knowledge-Guided": The authors describe their method as "knowledge-guided" only because they use prior knowledge as node features. However, it might be misleading as the utilization of prior knowledge as node features is relatively common. To justify the use of the term "knowledge-guided", the authors could provide more compelling reasons or evidence for why their use of prior knowledge goes beyond common practices in the field.

Response: We apologize for any confusion caused by the previous description regarding how we incorporate additional knowledge into our pre-training framework. In the field of molecular graph modeling using deep learning models, it is indeed a common practice to initialize nodes with atom-specific features such as atom type, degree, and formal charge. In our proposed method, the nodes were also initialized using these atom features, as detailed in Supplementary Table 1. However, it is crucial to clarify that the “additional knowledge” mentioned in our knowledge-guided pre-training strategy does not refer to these atom features. As explained in the subsection titled “Overview of KPGT” and the Methods section of our manuscript, we introduced a knowledge node (K node) in each molecular graph, connecting it to each individual node. While the common nodes (i.e., atoms) in the molecular graph are initialized with atom features, this K node is initialized with additional knowledge that quantitatively describes the characteristics of molecules. Specifically, we employed molecular descriptors and fingerprints as additional knowledge in our framework. To the best of our knowledge, KPGT is the “first” model to utilize such additional knowledge to guide the pre-training on molecules. The key distinction between atom features and molecular descriptors or fingerprints is that atom features describe atom-level features, while molecular descriptors and fingerprints capture characteristics at the molecule level. Molecular descriptors are typically generated by logical and mathematical procedures to quantitatively describe the physical and chemical profiles of molecules. For example, molecular weight and LogP (partition coefficient) are examples of molecular descriptors that provide information about the average molecular weight and the log of the partition coefficient of a solute between octanol and water, respectively. On the other hand, molecular fingerprints are binary bits that indicate the presence or absence of specific substructures within a molecule (examples of molecular fingerprints are illustrated in Fig. R22). Therefore, our proposed method stands apart from previous baseline methods through the integration of such additional knowledge.

Next, we would like to clarify the significance of incorporating additional knowledge into our pre-training process for molecular graphs. As discussed in the Introduction section, current self-supervised learning-based methods on molecular graphs typically rely on masking a proportion of atoms or substructures. However, molecular graphs possess semantics that are intricately tied to their structures. Even a small modification to the structure of a molecular graph, such as removing a single node, can lead to significant changes in its characteristics, resulting in the loss of inherent semantic information (Supplementary Fig. 1). This limitation potentially restricts current self-supervised learning-based methods on molecular graphs to capturing structural similarity among molecules, and thus fails to capture the rich semantic information related to molecular properties encoded in their chemical structures. Furthermore, without their inherent semantic information, the only dependency between the masked nodes and their adjacent nodes is the valency rules, which often fail to guide the model in making accurate predictions for the masked nodes, leading to the potential issue that the model simply memorizes the whole dataset (Supplementary Fig. 2). Therefore, we utilize the additional knowledge to initialize the K node, which basically serves as a valuable source of semantic information. Such incorporation of additional knowledge effectively guides the model in making accurate predictions for the masked nodes and capturing the intricate relationships between molecular structures and semantics. In our computational tests, KPGT consistently achieved superior prediction performance compared to baseline methods on molecular property prediction tasks across various domains (elaborated in the Results section). In addition, the ablation studies demonstrated that the prediction performance was degraded when additional knowledge was not incorporated into the pre-training process (Table R5). These results

further reinforced our point that incorporating additional knowledge is essential for improving the self-supervised learning results on molecular graphs.

Fig. R22: An example of RDKit fingerprints of Epinephrine. The numbers below the substructures indicate the positions of the corresponding fingerprints in the bit string.

10. **Address Pre-training Time Concerns:** It is mentioned that the pre-training process takes a long time, i.e., about 2 days. This extended duration may pose a challenge for nonexperts or users with limited computational resources. It would be beneficial to address this concern by discussing potential strategies for reducing the pre-training time or providing guidance on how to handle such computational requirements.

Response: Pre-training models with a large number of parameters on large-scale datasets is known to be a time-consuming process. For instance, the baseline methods GROVER¹⁹ and MolFormer⁵⁹ require 4 and 8 days of pre-training, respectively. This time requirement makes it impractical for users with limited computational resources to pre-train the KPGT used in our manuscript. However, the typical usage of a pre-trained model is to directly finetune it on downstream tasks, which generally requires significantly less time and computational resources in comparison with the pre-training process. We have measured the execution time of finetuning the pre-trained KPGT on a dataset consisting of approximately 4,200 molecules. The process took approximately 5 minutes on a single Nvidia A100 GPU and 11 minutes on a single Nvidia 1080TI GPU. Therefore, we believe that finetuning the pre-trained KPGT is user-friendly and accessible, even for users with only limited computational resources.

Moreover, to facilitate the utilization of our work by the research community, we have made the source code and pre-trained model used in our computational tests available on GitHub. The repository can be accessed at the following link: <https://github.com/lihan97/KPGT>. In addition, we have provided a user-friendly README file that offers detailed instructions on how to perform finetuning with our pre-trained model on a molecular property dataset of interest. By providing access to the codes and pre-trained model, we thus eliminate the need for users to conduct the entire pre-training process themselves. Instead, they can directly finetune the provided model on

the molecular property dataset of interest.

11. Provide Supporting Evidence: “Such backbone models generally provide low model capacity and thus potentially fail to capture the wide range of information required for the prediction of various properties for a diversity of molecules.” The description might not be accurate. To support this claim, it would be advantageous to cite relevant literature that demonstrates the limitations of previous models in capturing diverse molecular properties.

Response: We thank the reviewer for the nice suggestion. We agree with the reviewer's perspective that substantiating our claim regarding the limitation of graph neural networks (GNNs) with relevant literature is imperative. It is widely recognized that the chemical space is vast, probably with more than 10^{60} small molecules that hold potential for drug discovery⁶⁰⁻⁶². Additionally, molecular properties encompass a wide range of categories, including absorption, distribution, metabolism, excretion, toxicity, and binding affinity against various targets^{26,34,53,63}. To effectively model these diverse properties across a broad spectrum of molecules, a high-capacity model with a sufficient number of trainable parameters is crucial. Previous approaches for self-supervised learning on molecules typically employed GNNs as the underlying models. However, it is well known that GNNs suffer from over-smoothing, wherein node representations become increasingly similar and the model performance on downstream tasks significantly degrades as the number of layers increases^{64,65}. This limitation restricts the model capacity of GNNs. Also, GNNs typically rely on exchanging information between one-hop neighbors to construct node representations at each layer. In principle, such networks are unable to capture long-range interactions between atoms, which may be necessary or desired for accurate prediction of molecular properties^{66,67}. We have revised our manuscript to include citations of relevant literature that support our claim. We present the relevant paragraph in the Introduction section (Lines 70-77, Page 3) below for review:

“Existing self-supervised learning methods generally rely on GNNs (e.g., graph isomorphism network⁴⁵) as backbone models. However, GNNs can only provide limited model capacity, as they suffer from over-smoothing when increasing their number of layers^{46,47}. Additionally, GNNs might struggle to capture long-range interactions between atoms, arising from their standard practice of exchanging information only among one-hop neighbors during updates^{48,49}. Recent advancements in backbone networks, particularly transformer-based models⁵⁰, have emerged as game-changers⁵⁰⁻⁵². These models, characterized by an increasing number of parameters and the capability to capture long-range interactions, present promising avenues to comprehensively model the structural characteristics of molecules⁵³⁻⁵⁷.”

12. Update Figure 1: The legend of Figure 1 mentions abbreviations, but the DE and PE modules are not reflected in the figure. This inconsistency should be addressed to ensure clarity.

Response: We apologize for the lack of clarity in Fig. 1. To ensure a clear presentation, we have revised this figure to make the DE and PE modules easier to locate. The revised Fig. 1 has been attached below for review (Fig. R23). The modifications in the figure have been highlighted using dashed red box.

13. Make Code Accessible: Although the authors provide a code link, it is currently inaccessible, making it difficult to validate the reproducibility of the reported results. The authors are suggested to

Fig. R23: An illustrative diagram of KPGT. **a** A knowledge-guided pre-training strategy based on a masked graph model and enhanced by additional knowledge. Molecules are represented as molecular line graphs, which represent the adjacencies between the edges of the original molecular graphs. **b** A line graph transformer based on classic transformer architecture. **c** Transfer learning for downstream molecular property prediction under the finetuning setting, where the parameters of the pre-trained LiGhT are trainable. Various finetuning strategies, such as layer-wise learning rate decay, re-initialization, FLAG, and L^2 -SP are introduced in this setting. **d** Transfer learning for downstream molecular property prediction under the feature extraction setting, where the parameters of the pre-trained LiGhT are fixed. The neural fingerprints represent the molecular feature representations generated by the pre-trained LiGhT, serving as informative and discriminative representations of molecules. Abbreviations: multi-layer perceptron (MLP), linear layer (Linear), matrix multiplication (MatMul), distance encoding (DE) module, path encoding (PE) module.

make the code publicly accessible.

Response: We thank the reviewer for pointing this out. We fully concur with the reviewer's insights on the significance of providing accessible code for the sake of the reproducibility of our results. In response to this valuable suggestion, we have taken the necessary steps to ensure the availability of our code and datasets on GitHub. These resources are now accessible through the following link: <https://github.com/lihan97/KPGT>. To guarantee the ease of utilization, we have made a comprehensive README file that furnishes users with detailed instructions for executing the pre-training and finetuning processes. Moreover, we have shared the pre-trained model utilized in the computational tests presented in our manuscript. By providing these resources, we aim to facilitate not only the reproduction of our results on downstream tasks but also to empower users to apply the pre-trained model to their specific molecular property prediction tasks. To reflect these changes, we have revised the Code availability section (Lines 518-519, Page 18) as follows:

"Our source code and pre-trained model are available on GitHub, which can be accessed through the following link: <https://github.com/lihan97/KPGT>."

Reviewer 3

1. Remarks to the Author: In this paper, Li and others proposed KPGT, a novel knowledge-guided transformer model. This method is an integration of graph transformer and a pre-training strategy. It has been used for molecular property prediction (compared to 55 baselines on 63 datasets), and identify anti-tumor inhibitors. Overall, KPGT presents a novel pre-training framework, which could be a game changer for drug discovery. The extensive results also confirm the effectiveness of KPGT. I have a few comments for this manuscript to further improve and clarify its contribution.

Response: We appreciate the reviewer's positive feedback on our proposed method. We have carefully considered the reviewer's comments and suggestions to further improve and clarify the contribution of our work. Below, we will describe in detail how we addressed each of the reviewer's comments.

2. Section 2.2 "The mean absolute error (MAE) and Spearman's correlation coefficient (Spearman's r) were employed to evaluate the prediction performance of KPGT on regression tasks, while AUROC and the area under the precision-recall curve (AUPRC) were used to evaluate the prediction performance of KPGT on classification tasks", where are the results of AUPRC and correlation? I didn't find it in the main figures or supplementary figures.

Response: We apologize for the ambiguity made in our original description of the evaluation metrics used in the tests on the datasets from the TDC benchmarking platform^{53,54}. In our experiments, we strictly adhered to the evaluation protocol provided by TDC^{53,54}. It is important to note that each dataset employed only one metric for evaluation, following the evaluation protocol. For example, we utilized AUROC as the metric for evaluating the prediction performance on the HIA dataset, and AUPRC was used for the CYP2C19_{inh} dataset. We have revised the original sentence "The mean absolute error (MAE) and Spearman's correlation coefficient (Spearman's r) were employed to evaluate the prediction performance of KPGT on regression tasks, while AUROC and the area under the precision-recall curve (AUPRC) were used to evaluate the prediction performance of KPGT on classification tasks" to "The mean absolute error (MAE) or Spearman's correlation coefficient (Spearman's r) was employed to evaluate the prediction performance of KPGT on regression tasks, while AUROC or the area under the precision-recall curve (AUPRC) was used to evaluate the prediction performance of KPGT on classification tasks". Additionally, a comprehensive overview of the evaluation metrics used for all the datasets in TDC can be found in Table R6.

3. Please release the code and data for the 55 baselines and how to reproduce their results from the 63 datasets.

Response: We appreciate the valuable suggestion from the reviewer. In the Results section of our manuscript, we conducted a comprehensive evaluation on the prediction performance of KPGT by comparing it with a number of baseline methods on sixty-three datasets in our benchmarking tests. In our first benchmarking test, we compared KPGT with nineteen self-supervised learning-based methods (with five newly incorporated self-supervised learning-based baseline methods) on eleven datasets that had been widely used for evaluating self-supervised learning-based methods in molecular property prediction tasks. To ensure a fair comparison, we directly utilized the pre-trained models released by the corresponding authors on their corresponding

Table R6: Prediction performance on the datasets from the TDC benchmark of KPGT and the best baseline methods provided by the leaderboards of the TDC benchmark. The numbers in brackets are the standard deviations over five independent runs. The best result for each dataset is marked in bold. Abbreviations: deep learning based method (DL), machine learning based method (ML).

Group	Dataset description			Current best baseline			KPGT	
	Dataset	Task	Metric	Method	Type	Score	Score	Rank
Absorption	Caco2	Regression	MAE	BaseBoosting	ML	0.285(0.005)	0.284 (0.009)	1st
	HIA	Classification	AUROC	RFStacker	ML	0.988 (0.002)	0.982(0.004)	2nd
	Pgp	Classification	AUROC	ZairaChem	ML	0.935(0.006)	0.938 (0.004)	1st
	Bioav	Classification	AUROC	SimGCN	DL	0.748(0.033)	0.750 (0.022)	1st
	Lipo	Regression	MAE	Chemprop-RDKit	DL	0.466(0.006)	0.446 (0.016)	1st
	AqSol	Regression	MAE	Chemprop-RDKit	DL	0.762(0.004)	0.714 (0.011)	1st
Distribution	BBB	Classification	AUROC	LRE	DL	0.962 (0.003)	0.908(0.005)	6th
	PPBR	Regression	MAE	Chemprop	DL	7.811(0.163)	7.684 (0.250)	1st
	VDss	Regression	Spearman's r	Basic ML	ML	0.627(0.010)	0.633 (0.016)	1st
Metabolism	CYP2C9 _{inh}	Classification	AUPRC	ZairaChem	ML	0.786(0.004)	0.797 (0.006)	1st
	CYP2D6 _{inh}	Classification	AUPRC	Chemprop-RDKit	DL	0.672(0.008)	0.724 (0.008)	1st
	CYP3A4 _{inh}	Classification	AUPRC	ZairaChem	ML	0.875(0.002)	0.894 (0.004)	1st
	CYP2C9 _{sub}	Classification	AUPRC	ZairaChem	ML	0.441(0.033)	0.450 (0.044)	1st
	CYP2D6 _{sub}	Classification	AUPRC	Contextpred	DL	0.736(0.024)	0.737 (0.016)	1st
	CYP3A4 _{sub}	Classification	AUPRC	GNN	DL	0.662(0.031)	0.730 (0.023)	1st
Excretion	HalfLife	Regression	Spearman's r	Euclia ML model	ML	0.547 (0.032)	0.531(0.030)	3rd
	CL-Hepa	Regression	Spearman's r	Basic ML	ML	0.440 (0.003)	0.424(0.019)	6th
	CL-Micro	Regression	Spearman's r	RFStacker	ML	0.625(0.002)	0.637 (0.010)	1st
Toxicity	LD50	Regression	MAE	BaseBoosting	ML	0.552(0.009)	0.545 (0.010)	1st
	hERG	Classification	AUROC	SimGCN	DL	0.874 (0.014)	0.847(0.024)	3rd
	Ames	Classification	AUROC	ZairaChem	ML	0.871 (0.002)	0.868(0.003)	2nd
	DILI	Classification	AUROC	ZairaChem	ML	0.925(0.005)	0.929 (0.013)	1st

GitHub repositories. We finetuned these pre-trained models on each dataset using the provided scripts from the GitHub repositories. To facilitate the access to the baseline methods, we have collected the GitHub repositories of these methods and released the compiled codes on our own GitHub repository, which can be accessed at <https://github.com/lihan97/KPGT>. Regarding the tests on the datasets from the TDC benchmarking platform^{53,54}, we obtained the prediction performance of each method on each dataset from the leaderboards available on the TDC website. The leaderboards can be accessed at https://tdcommons.ai/benchmark/admet_group/01caco2/. To reproduce the results of individual baseline methods, users can refer to the GitHub repositories provided on the leaderboards. For the evaluation on the datasets from the MoleculeACE platform²⁶, we acquired the results of baseline methods from its GitHub repository. Specifically, we retrieved the data from the file located at https://github.com/molML/MoleculeACE/blob/main/MoleculeACE/Data/results/MoleculeACE_results.csv. The reproduction of results can be achieved by following the instructions provided in the README file of the GitHub repository.

Furthermore, we have uploaded our codes and all the datasets from these three benchmarks on GitHub at the following link: <https://github.com/lihan97/KPGT>. Additionally, we have made the pre-trained model used in the computational tests presented in our manuscript available to users. We have also provided a comprehensive README file which includes detailed instructions for executing the finetuning process with the pre-trained model, enabling the easy replication of results on the downstream tasks and the application of the pre-trained model to other molecular property prediction tasks of interest. We have revised our Code availability (Lines 518-519, Page 18) as follows: “Our source code and pre-trained model are available on GitHub, which can be accessed through the following link: <https://github.com/lihan97/KPGT>.”

4. Please clarify why most of the baselines used in figure 2 are not compared in the experiment in figure 4. Are they not applicable in the time splitting and domain transfer settings?

Response: We thank the reviewer for bringing up this question. The reason for not including all the baseline methods in Fig. 4 is that the benchmarking tests presented in the subsection named “KPGT outperforms the baseline methods on molecular property prediction” within the Results section already provided a comprehensive comparison of KPGT with the baseline methods on the molecular property prediction tasks. The results from these tests had adequately demonstrated the superiority of KPGT in predicting molecular properties over baseline methods. To avoid the duplication of the evaluation already presented in the Results section, we chose to include two representative self-supervised learning methods, namely GROVER¹⁹ and GraphCL¹⁶, as baseline methods in this test. To further evaluate the prediction performance of KPGT on predicting inhibitors for the specific antitumor targets, namely HPK1 and FGFR1, we have conducted additional tests to incorporate all the nineteen self-supervised learning-based baseline methods from the first benchmarking test in the Results section. Fig. R24 reports the comparative results on HPK1 dataset, which demonstrated that KPGT significantly outperformed nineteen self-supervised learning-based baseline methods in terms of both Spearman’s r and Pearson’s r . Fig. R25 presents the test results on FGFR1 dataset. From the results, KPGT achieved higher correlation values under both the scaffold splitting and the time splitting scenarios in comparison with baseline methods. It is pertinent to note that during these tests, the molecules within the HPK1 and FGFR1 datasets that overlapped with the pre-training data employed by KPGT were omitted (as elaborated in our response to Point 7). We have updated the results to Figs. 4-5 and revised the relevant paragraphs in the subsection titled “Uncovering potentially effective

inhibitors for antitumor targets by KPGT” within the Results section (Lines 254-261, Page 11 and Line 291-293, Page 13) as follows:

“Comparison results between KPGT and nineteen self-supervised learning-based baseline methods are detailed in Figs. 4b-d and Supplementary Fig. 14. The results demonstrated that KPGT significantly outperformed nineteen self-supervised learning-based baseline methods in terms of Spearman’s r and Pearson’s r . Remarkably, even in the time splitting and domain transfer scenarios, where the molecules within training and test sets were significantly different in their structures (Fig. 4a), KPGT consistently achieved elevated correlation scores. To elaborate, KPGT achieved Pearson’s r scores of 0.731 and 0.805 under the time splitting and domain transfer scenarios, respectively. These observations validated the superior generalizability and reliability of KPGT in the prediction of HPK1 inhibitors.”

“Fig. 5a-b and Supplementary Fig. 15 illustrate the performance of KPGT and nineteen self-supervised learning-based baseline methods. KPGT achieved high correlation values under both the scaffold splitting (Pearson’s $r = 0.924$) and the time splitting (Pearson’s $r = 0.716$) scenarios.”

Fig. R24: Prediction performance of KPGT and baseline methods on the HPK1 dataset under the scaffold splitting, time splitting, and domain transfer scenarios, measured in terms of both Spearman’s r and Pearson’s r .

- The drug molecular graphs showed in figure 4e are not informative. Do drug structures reflect potency?

Fig. R25: Prediction performance of KPGT and baseline methods on the FGFR1 dataset under the scaffold splitting and time splitting scenarios, measured in terms of both Spearman's r and Pearson's r.

Response: We appreciate the valuable feedback provided by the reviewer regarding Fig. 4e. We agree with the reviewer that the information provided in the previous version of Fig. 4e was limited, as it only listed the top ten molecules identified by KPGT that potentially inhibited HPK1. To address this concern, we have made the corresponding revisions to Fig. 4e. In particular in the updated versions of Fig. 4e, we have added check symbols next to the molecules that had been identified as inhibitors of HPK1 in the literature. As depicted in Fig. R26a, nine out of the top ten molecules identified by KPGT has literature evidence supporting their inhibitory activity against HPK1, indicating the superior reliability of KPGT in identifying potential inhibitors for HPK1.

Additionally, in our tests on identifying inhibitors for antitumor targets using KPGT, we have carried out docking analyses on the molecules identified by KPGT that potentially inhibit HPK1 and FGFR1 to strengthen our findings (Figure R26 and Figure R27). We present the relevant paragraphs in the subsection named "Ablation studies on KPGT" within the Results section (Lines 271-285, Page 11 and Lines 296-304, Page 13) below for review:

"To strengthen our findings, we further conducted docking analyses for the top twenty predictions from KPGT. Autodock Vina^{71,72}, a widely used docking software, was employed for these tests. The reference protein-ligand structure (PDB ID: 7SIU⁷⁴) guided the identification of the binding pocket. As depicted in Fig. 4f, all the molecules achieved docking energies below -7 kcal/mol, a commonly used threshold for drug-like molecules⁸²⁻⁸⁵, signifying substantial potential for these molecules as HPK1 inhibitors. Additionally, we conducted an in-depth analysis of the protein-ligand interactions for all the molecules that had not been reported in the literature using a widely applied protein-ligand interaction profiler named PLIP⁷³. Fig. 4g illustrates the protein-ligand interaction profile of the ligand gilteritrib with the protein HPK1. The analysis revealed the formation of three hydrophobic interactions and six hydrogen bonds between the ligand and the protein. Remarkably, the hydrogen bonds formed with residues 94A and 97A were also reported in the reference protein-ligand structure (PDB ID: 7SIU⁷⁴). These observations showcased that the

molecules can tightly bind to HPK1, validating the reliability of the docking results. Supplementary Fig. 16 provides additional protein-ligand interaction profiles for other molecules, including palbociclib, ripretinib, trilaciclib, rucaparib, selpercatinib, atrofloxacin, and vericiguat. Collectively, these results highlighted the superior ability of KPGT in identifying potential inhibitors for HPK1.”

“For the docking tests, the protein-ligand structure (PDB ID: 5A4C⁸¹) was employed as a reference for the binding pocket identification. As shown in Fig. 5d, all the top twenty molecules identified by KPGT achieved docking energies below -7 kcal/mol. By profiling the protein-ligand interactions utilizing PLIP⁷³, the ligand brigatinib tightly bound to the protein FGR1. Specifically, it formed four hydrophobic interactions, one hydrogen bond, and one salt bridge with FGFR1 (Fig. 5e). Among these interactions, the hydrogen bond formed with residue 641A was also reported in the reference protein-ligand structure (PDB ID: 5A4C⁸¹). In Supplementary Fig. 17, we also display additional protein-ligand interaction profiles for other molecules, including, ripretinib, encorafenib, elagolix, baricitinib, enasidenib, and ruxolitinib.”

Fig. R26: Identifying inhibitors against HPK1 using KPGT. **a** Visualization of molecular representations of molecules from the HPK1 pIC50 dataset and the FDA dataset derived from KPGT. The top ten predictions of potential inhibitors against HPK1 from the FDA dataset derived from KPGT are delineated in dashed circles, and their corresponding molecular structures are listed in the right panel. The check symbols indicate that the molecules had been previously identified as inhibitors of HPK1 in previous studies. **b** The docking scores of the top twenty inhibitors against HPK1 identified by KPGT, measured by Autodock Vina^{36,37}. **c** The interactions between Giliteritnib and HPK1 profiled by PLIP⁴³. The protein-ligand structure (PDB ID: 7siu³⁸) was utilized as a reference for the identification of the binding pocket. The read line at -7 kcal/mol represents a commonly used threshold for identifying drug-like molecules.

- It might be interesting to perform graph-based explanation approach (GNExplainer) to understand the important atoms and sub-structures in each graph in figure 4e.

Fig. R27: Identifying inhibitors against HPK1 using KPGT. **a** Visualization of molecular representations of molecules from the pIC50 dataset of FGFR1 inhibitors and FDA dataset derived from KPGT. The top ten predictions of potential inhibitors against FGFR1 from the FDA dataset derived from KPGT were delineated in dashed circles, and their corresponding structures were listed in the right panel. The check symbols indicate that the molecules had been previously identified as inhibitors of FGFR1 in previous studies. **b** The docking scores of the top twenty potential inhibitors identified by KPGT, measured by Autodock Vina^{36,37}. **c** The interactions between Brigatinib and FGFR1 profiled by PLIP⁴³. The protein-ligand structure (PDB ID: 5a4c⁴⁴) was utilized as a reference for the identification of the binding pocket. The read line at -7 kcal/mol represents a commonly used threshold for identifying drug-like molecules.

Response: We sincerely appreciate the reviewer's insightful suggestion. We concur that employing a graph-based explanation approach to provide insights into the molecules potentially inhibiting HPK1, as identified by KPGT, is indeed an interesting avenue of exploration. In response to the reviewer's recommendation, we have integrated SubgraphX⁶⁸, a recently introduced graph-based explanation approach, into our analysis. Our initial assessment focused on evaluating the explanation capability of SubgraphX by replicating the tests conducted in Fig. 3d within the Results section. As presented in Figs. R28- R29, the results indicated that SubgraphX more accurately identifies the distinguishing substructures responsible for activity cliffs in comparison with Gradient*Input⁶⁹. This outcome underscored the superior explanatory prowess of SubgraphX compared to Gradient*Input. Next, we employed SubgraphX to gain insights into the important atoms within the molecules identified by KPGT as potential HPK1 inhibitors. Fig. R30 illustrates the results of this analysis. The important nodes identified by SubgraphX overlapped with the atoms involved in the interactions with the HPK1 protein profiled by PLIP⁴³. For example, SubgraphX identified three atoms that form interactions with HPK1. These observations not only demonstrated the exceptional explanatory capability of SubgraphX but also affirmed that KPGT can offer valuable interpretability for its predictions. We have revised Fig. 3d to include the results

obtained using SubgraphX and incorporated the explanation results on the potential inhibitors against HPK1 identified by KPGT into Supplementary Fig. 18.

Fig. R28: SubgraphX⁶⁸ identifies atoms distinguishing activity cliffs.

Fig. R29: Gradient*Input⁶⁹ identifies atoms distinguishing activity cliffs.

7. For the 4442 molecules identified for validating HPK1, are they excluded from the pretrained data used by KPGT?

Response: We appreciate the reviewer for bringing up this important point. It is crucial to validate that the molecules present in the HPK1 and FGFR1 datasets do not intersect with the data used for pre-training, to ensure the integrity of our comparisons and prevent any potential data leakage. Upon further analysis, we have confirmed the presence of molecules within both the HPK1 and FGFR1 datasets that overlap with the pre-training data employed by KPGT. Specifi-

Fig. R30: SubgraphX⁶⁸ identifies atoms that form interactions with HPK1 profiles by PLIP⁴³.

cally, we have identified 55 molecules within the HPK1 dataset and 2,709 molecules within the FGFR1 dataset that exhibit such overlap. As a corrective measure, we have excluded these overlapping molecules and conducted the prediction tests again. The results of these tests have been elaborated in our response to Point 4. In light of these adjustments, we have revised the related descriptions regarding the datasets acquisition in the subsection titled “Uncovering potentially effective inhibitors for antitumor targets by KPGT” within the Results section (Lines 252-254 and Lines 290-291, Page 11) as follows:

“It’s important to note that we excluded the molecules that overlapped with the pre-training data employed by KPGT in this test.”

“Notably, we omitted the molecules within this dataset that overlapped with the pre-training data employed in this test.”

References

1. Han Van De Waterbeemd and Eric Gifford. Admet in silico modelling: towards prediction paradise? *Nature reviews Drug discovery*, 2(3):192–204, 2003.
2. Jie Dong, Ning-Ning Wang, Zhi-Jiang Yao, Lin Zhang, Yan Cheng, Defang Ouyang, Ai-Ping Lu, and Dong-Sheng Cao. Admetlab: a platform for systematic admet evaluation based on a comprehensively collected admet database. *Journal of cheminformatics*, 10(1):1–11, 2018.
3. Keith T Butler, Daniel W Davies, Hugh Cartwright, Olexandr Isayev, and Aron Walsh. Machine learning for molecular and materials science. *Nature*, 559(7715):547–555, 2018.
4. Zheng Xu, Sheng Wang, Feiyun Zhu, and Junzhou Huang. Seq2seq fingerprint: An unsupervised deep molecular embedding for drug discovery. In *Proceedings of the 8th ACM international conference on bioinformatics, computational biology, and health informatics*, pages 285–294, 2017.
5. Zhe Quan, Xuan Lin, Zhi-Jie Wang, Yan Liu, Fan Wang, and Kenli Li. A system for learning atoms based on long short-term memory recurrent neural networks. In *2018 IEEE International Conference on Bioinformatics and Biomedicine (BIBM)*, pages 728–733. IEEE, 2018.
6. Esben Jannik Bjerrum. Smiles enumeration as data augmentation for neural network modeling of molecules. *arXiv preprint arXiv:1703.07076*, 2017.
7. Tingting Shi, Yingwu Yang, Shuheng Huang, Linxin Chen, Zuyin Kuang, Yu Heng, and Hu Mei. Molecular image-based convolutional neural network for the prediction of admet properties. *Chemometrics and Intelligent Laboratory Systems*, 194:103853, 2019.
8. Yasunari Matsuzaka and Yoshihiro Uesawa. Optimization of a deep-learning method based on the classification of images generated by parameterized deep snap a novel molecular-image-input technique for quantitative structure–activity relationship (qsar) analysis. *Frontiers in bioengineering and biotechnology*, 7:65, 2019.
9. Atsushi Yoshimori. Prediction of molecular properties using molecular topographic map. *Molecules*, 26(15):4475, 2021.
10. Justin Gilmer, Samuel S Schoenholz, Patrick F Riley, Oriol Vinyals, and George E Dahl. Neural message passing for quantum chemistry. In *International conference on machine learning*, pages 1263–1272. PMLR, 2017.
11. Zhaoping Xiong et al. Pushing the boundaries of molecular representation for drug discovery with the graph attention mechanism. *Journal of medicinal chemistry*, 63(16):8749–8760, 2019.
12. Gabriele Corso, Luca Cavalleri, Dominique Beaini, Pietro Liò, and Petar Velickovic. Principal neighbourhood aggregation for graph nets. In *NeurIPS 2020*, 2020.
13. Dominique Beaini, Saro Passaro, Vincent Létourneau, Will Hamilton, Gabriele Corso, and Pietro Liò. Directional graph networks. In *International Conference on Machine Learning*, pages 748–758. PMLR, 2021.
14. Weihua Hu, Bowen Liu, Joseph Gomes, Marinka Zitnik, Percy Liang, Vijay S. Pande, and Jure Leskovec. Strategies for pre-training graph neural networks. In *ICLR 2020*, 2020.
15. Shengchao Liu, Mehmet Furkan Demirel, and Yingyu Liang. N-gram graph: Simple unsupervised representation for graphs, with applications to molecules. In *NeurIPS 2019*, pages 8464–8476, 2019.
16. Yuning You, Tianlong Chen, Yongduo Sui, Ting Chen, Zhangyang Wang, and Yang Shen. Graph contrastive learning with augmentations. In *NeurIPS 2020*, 2020.
17. Minghao Xu, Hang Wang, Bingbing Ni, Hongyu Guo, and Jian Tang. Self-supervised graph-level representation learning with local and global structure. In *ICML 2021*, volume 139, pages 11548–11558, 2021.
18. Yuning You, Tianlong Chen, Yang Shen, and Zhangyang Wang. Graph contrastive learning automated. In *ICML 2021*, volume 139, pages 12121–12132, 2021.
19. Yu Rong, Yatao Bian, Tingyang Xu, Weiyang Xie, Ying Wei, Wenbing Huang, and Junzhou Huang. Self-supervised graph transformer on large-scale molecular data. In Hugo Larochelle, Marc’Aurelio Ranzato, Raia Hadsell, Maria-Florina Balcan, and Hsuan-Tien Lin, editors, *Advances in Neural Information Processing Systems 33: Annual Conference on Neural Information Processing Systems 2020, NeurIPS 2020, December 6-12, 2020, virtual*, 2020.

20. Yuyang Wang, Jianren Wang, Zhonglin Cao, and Amir Barati Farimani. Molclr: molecular contrastive learning of representations via graph neural networks. arXiv preprint arXiv:2102.10056, 2021.
21. Hannes Stärk, Dominique Beaini, Gabriele Corso, Prudencio Tossou, Christian Dallago, Stephan Günemann, and Pietro Liò. 3d infomax improves gnns for molecular property prediction. arXiv preprint arXiv:2110.04126, 2021.
22. Shengchao Liu, Hanchen Wang, Weiyang Liu, Joan Lasenby, Hongyu Guo, and Jian Tang. Pre-training molecular graph representation with 3d geometry. arXiv preprint arXiv:2110.07728, 2021.
23. Xiaomin Fang, Lihang Liu, Jieqiong Lei, Donglong He, Shanzhuo Zhang, Jingbo Zhou, Fan Wang, Hua Wu, and Haifeng Wang. Geometry-enhanced molecular representation learning for property prediction. *Nature Machine Intelligence*, 4(2):127–134, 2022.
24. Zhenyu Hou, Xiao Liu, Yukuo Cen, Yuxiao Dong, Hongxia Yang, Chunjie Wang, and Jie Tang. Graphmae: Self-supervised masked graph autoencoders. In Proceedings of the 28th ACM SIGKDD Conference on Knowledge Discovery and Data Mining, pages 594–604, 2022.
25. Jun Xia, Chengshuai Zhao, Bozhen Hu, Zhangyang Gao, Cheng Tan, Yue Liu, Siyuan Li, and Stan Z Li. Molebert: Rethinking pre-training graph neural networks for molecules. In The Eleventh International Conference on Learning Representations, 2022.
26. Derek van Tilborg, Alisa Alenicheva, and Francesca Grisoni. Exposing the limitations of molecular machine learning with activity cliffs. *Journal of Chemical Information and Modeling*, 62(23):5938–5951, 2022. PMID: 36456532.
27. Chengxuan Ying, Tianle Cai, Shengjie Luo, Shuxin Zheng, Guolin Ke, Di He, Yanming Shen, and Tie-Yan Liu. Do transformers really perform badly for graph representation? In NeurIPS 2021, 2021.
28. Keyulu Xu, Weihua Hu, Jure Leskovec, and Stefanie Jegelka. How powerful are graph neural networks? In ICLR 2019, 2019.
29. Ashish Vaswani, Noam Shazeer, Niki Parmar, Jakob Uszkoreit, Llion Jones, Aidan N. Gomez, Lukasz Kaiser, and Illia Polosukhin. Attention is all you need. In NeurIPS 2017, pages 5998–6008, 2017.
30. Laurens Van der Maaten and Geoffrey Hinton. Visualizing data using t-sne. *Journal of machine learning research*, 9(11), 2008.
31. Henrike Veith, Noel Southall, Ruili Huang, Tim James, Darren Fayne, Natalia Artemenko, Min Shen, James Ingles, Christopher P Austin, David G Lloyd, et al. Comprehensive characterization of cytochrome p450 isozyme selectivity across chemical libraries. *Nature biotechnology*, 27(11):1050–1055, 2009.
32. David Rogers and Mathew Hahn. Extended-connectivity fingerprints. *Journal of chemical information and modeling*, 50(5):742–754, 2010.
33. Landrum Greg et al. rdkit/rdkit: 2021_09_2 (q3 2021) release, October 2021.
34. Zhenqin Wu, Bharath Ramsundar, Evan N Feinberg, Joseph Gomes, Caleb Geniesse, Aneesh S Pappu, Karl Leswing, and Vijay Pande. Moleculenet: a benchmark for molecular machine learning. *Chemical science*, 9(2):513–530, 2018.
35. Weihua Hu, Matthias Fey, Marinka Zitnik, Yuxiao Dong, Hongyu Ren, Bowen Liu, Michele Catasta, and Jure Leskovec. Open graph benchmark: Datasets for machine learning on graphs. *Advances in neural information processing systems*, 33:22118–22133, 2020.
36. Jerome Eberhardt, Diogo Santos-Martins, Andreas F Tillack, and Stefano Forli. Autodock vina 1.2. 0: New docking methods, expanded force field, and python bindings. *Journal of chemical information and modeling*, 61(8):3891–3898, 2021.
37. Oleg Trott and Arthur J Olson. Autodock vina: improving the speed and accuracy of docking with a new scoring function, efficient optimization, and multithreading. *Journal of computational chemistry*, 31(2):455–461, 2010.
38. Sven Malchow, Alla Korepanova, Sanjay C Panchal, Ryan A McClure, Kenton L Longenecker, Wei Qiu, Hongyu Zhao, Min Cheng, Jun Guo, Kelly L Klinge, et al. The hpk1 inhibitor a-745 verifies the potential of modulating t cell kinase signaling for immunotherapy. *ACS Chemical Biology*, 17(3):556–566, 2022.
39. Max W Chang, William Lindstrom, Arthur J Olson, and Richard K Belew. Analysis of hiv wild-type and mutant structures via in silico docking against diverse ligand libraries. *Journal of chemical information and modeling*, 47(3):1258–1262, 2007.

40. Christopher Llynard D Ortiz, Gladys C Completo, Ruel C Nacario, and Ricky B Nellas. Potential inhibitors of galactofuranosyltransferase 2 (glft2): molecular docking, 3d-qsar, and in silico admetox studies. Scientific reports, 9(1):17096, 2019.
41. Sajjad Ahmad, Yasir Waheed, Asma Abro, Sumra Wajid Abbasi, and Saba Ismail. Molecular screening of glycyrrhizin-based inhibitors against ace2 host receptor of sars-cov-2. Journal of Molecular Modeling, 27(7):206, 2021.
42. Andrea Isabel Trujillo-Correa, Diana Carolina Quintero-Gil, Fredyc Diaz-Castillo, Winston Quiñones, Sara M Robledo, and Marlen Martinez-Gutierrez. In vitro and in silico anti-dengue activity of compounds obtained from psidium guajava through bioprospecting. BMC complementary and alternative medicine, 19:1–16, 2019.
43. Melissa F Adasme, Katja L Linnemann, Sarah Naomi Bolz, Florian Kaiser, Sebastian Salentin, V Joachim Haupt, and Michael Schroeder. Plip 2021: Expanding the scope of the protein–ligand interaction profiler to dna and rna. Nucleic acids research, 49(W1):W530–W534, 2021.
44. Tobias Klein, Navratna Vajpai, Jonathan J Phillips, Gareth Davies, Geoffrey A Holdgate, Chris Phillips, Julie A Tucker, Richard A Norman, Andrew D Scott, Daniel R Higazi, et al. Structural and dynamic insights into the energetics of activation loop rearrangement in fgfr1 kinase. Nature communications, 6(1):7877, 2015.
45. Corinna Cortes and Vladimir Vapnik. Support-vector networks. Machine learning, 20:273–297, 1995.
46. Jerome H Friedman. Greedy function approximation: a gradient boosting machine. Annals of statistics, pages 1189–1232, 2001.
47. Leo Breiman. Random forests. Machine learning, 45:5–32, 2001.
48. Ling Xue and Jurgen Bajorath. Molecular descriptors in chemoinformatics, computational combinatorial chemistry, and virtual screening. Combinatorial chemistry & high throughput screening, 3(5):363–372, 2000.
49. Adrià Cereto-Massagué et al. Molecular fingerprint similarity search in virtual screening. Methods, 71:58–63, 2015.
50. Andreas Mayr, Günter Klambauer, Thomas Unterthiner, Marvin Steijaert, Jörg K Wegner, Hugo Ceulemans, Djork-Arné Clevert, and Sepp Hochreiter. Large-scale comparison of machine learning methods for drug target prediction on chembl. Chemical science, 9(24):5441–5451, 2018.
51. Hirotomo Moriwaki, Yu-Shi Tian, Norihito Kawashita, and Tatsuya Takagi. Mordred: a molecular descriptor calculator. Journal of cheminformatics, 10(1):1–14, 2018.
52. Bharath Ramsundar, Peter Eastman, Patrick Walters, Vijay Pande, Karl Leswing, and Zhenqin Wu. Deep Learning for the Life Sciences. O'Reilly Media, 2019.
53. Kexin Huang, Tianfan Fu, Wenhao Gao, Yue Zhao, Yusuf Roohani, Jure Leskovec, Connor W Coley, Cao Xiao, Jimeng Sun, and Marinka Zitnik. Therapeutics data commons: Machine learning datasets and tasks for drug discovery and development. Proceedings of Neural Information Processing Systems, NeurIPS Datasets and Benchmarks, 2021.
54. Kexin Huang, Tianfan Fu, Wenhao Gao, Yue Zhao, Yusuf Roohani, Jure Leskovec, Connor W Coley, Cao Xiao, Jimeng Sun, and Marinka Zitnik. Artificial intelligence foundation for therapeutic science. Nature Chemical Biology, 2022.
55. Jr-Wen Shui, Jonathan S Boomer, Jin Han, Jun Xu, Gregory A Dement, Guisheng Zhou, and Tse-Hua Tan. Hematopoietic progenitor kinase 1 negatively regulates t cell receptor signaling and t cell–mediated immune responses. Nature immunology, 8(1):84–91, 2007.
56. Jingwen Si, Xiangjun Shi, Shuhao Sun, Bin Zou, Yaopeng Li, Dongjie An, Xingyu Lin, Yan Gao, Fei Long, Bo Pang, et al. Hematopoietic progenitor kinase1 (hpk1) mediates t cell dysfunction and is a druggable target for t cell-based immunotherapies. Cancer Cell, 38(4):551–566, 2020.
57. Victor D Acevedo, Rama D Gangula, Kevin W Freeman, Rile Li, Youngyou Zhang, Fen Wang, Gustavo E Ayala, Leif E Peterson, Michael Ittmann, and David M Spencer. Inducible fgfr-1 activation leads to irreversible prostate adenocarcinoma and an epithelial-to-mesenchymal transition. Cancer cell, 12(6):559–571, 2007.
58. PT Nguyen, T Tsunematsu, S Yanagisawa, Y Kudo, M Miyauchi, N Kamata, and T Takata. The fgfr1 inhibitor pd173074 induces mesenchymal–epithelial transition through the transcription factor ap-1. British journal of cancer, 109(8):2248–2258, 2013.

59. Jerret Ross, Brian Belgodere, Vijil Chenthamarakshan, Inkit Padhi, Youssef Mroueh, and Payel Das. Large-scale chemical language representations capture molecular structure and properties. Nature Machine Intelligence, 4(12):1256–1264, 2022.
60. Regine S Bohacek, Colin McMartin, and Wayne C Guida. The art and practice of structure-based drug design: a molecular modeling perspective. Medicinal research reviews, 16(1):3–50, 1996.
61. Peter Ertl. Cheminformatics analysis of organic substituents: identification of the most common substituents, calculation of substituent properties, and automatic identification of drug-like bioisosteric groups. Journal of chemical information and computer sciences, 43(2):374–380, 2003.
62. Jean-Louis Reymond, Ruud Van Deursen, Lorenz C Blum, and Lars Ruddigkeit. Chemical space as a source for new drugs. MedChemComm, 1(1):30–38, 2010.
63. Anna Gaulton et al. The chEMBL database in 2017. Nucleic acids research, 45(D1):D945–D954, 2017.
64. Deli Chen, Yankai Lin, Wei Li, Peng Li, Jie Zhou, and Xu Sun. Measuring and relieving the over-smoothing problem for graph neural networks from the topological view. In Proceedings of the AAAI conference on artificial intelligence, volume 34, pages 3438–3445, 2020.
65. Chen Cai and Yusu Wang. A note on over-smoothing for graph neural networks. arXiv preprint arXiv:2006.13318, 2020.
66. Vijay Prakash Dwivedi, Ladislav Rampásek, Michael Galkin, Ali Parviz, Guy Wolf, Anh Tuan Luu, and Dominique Beaini. Long range graph benchmark. Advances in Neural Information Processing Systems, 35:22326–22340, 2022.
67. Zhanghao Wu, Paras Jain, Matthew Wright, Azalia Mirhoseini, Joseph E Gonzalez, and Ion Stoica. Representing long-range context for graph neural networks with global attention. Advances in Neural Information Processing Systems, 34:13266–13279, 2021.
68. Hao Yuan, Haiyang Yu, Jie Wang, Kang Li, and Shuiwang Ji. On explainability of graph neural networks via subgraph explorations. In International conference on machine learning, pages 12241–12252. PMLR, 2021.
69. Avanti Shrikumar, Peyton Greenside, Anna Shcherbina, and Anshul Kundaje. Not just a black box: Learning important features through propagating activation differences. arXiv preprint arXiv:1605.01713, 2016.

Reviewers' Comments:

Reviewer #1:

Remarks to the Author:

The paper discusses the importance of learning effective molecular feature representations for drug discovery, particularly in the context of molecular property prediction. It highlights the recent trend of using self-supervised learning techniques to pre-train graph neural networks (GNNs) to address data scarcity challenges in this domain. Furthermore, it validates the practical utility of KPGT in drug discovery by successfully identifying potential inhibitors for two antitumor targets: hematopoietic progenitor kinase 1 (HPK1) and fibroblast growth factor receptor 1 (FGFR1).

The following are specific comments or suggestions:

1. I was interested in the difference between this paper and the publication in KDD 2022, KPGT: Knowledge-Guided Pre-training of Graph Transformer for Molecular Property Prediction. Discuss and comparison should be made.
2. The masked-based self-supervised molecular graph pre-training has been around for a while. The line graph introduced in this paper has already demonstrated its effectiveness at KDD 2022. Thus, the community's contributions to this paper are limited.

Reviewer #2:

Remarks to the Author:

The authors have carefully addressed most of the provided comments, while it is still expected to validate the generalization of the proposed model to different targets within the case study if possible. I recommend publishing this paper. Thank you very much!

Reviewer #3:

Remarks to the Author:

The authors have addressed all of my concerns.

Reviewer 1

1. Remarks to the Author: The paper discusses the importance of learning effective molecular feature representations for drug discovery, particularly in the context of molecular property prediction. It highlights the recent trend of using self-supervised learning techniques to pre-train graph neural networks (GNNs) to address data scarcity challenges in this domain. Furthermore, it validates the practical utility of KPGT in drug discovery by successfully identifying potential inhibitors for two antitumor targets: hematopoietic progenitor kinase 1 (HPK1) and fibroblast growth factor receptor 1 (FGFR1).

Response: We would like to thank the reviewer for the insightful comments in the first round that helped improve the quality of our manuscript as well as their further comments here.

2. I was interested in the difference between this paper and the publication in KDD 2022, KPGT: Knowledge-Guided Pre-training of Graph Transformer for Molecular Property Prediction. Discuss and comparison should be made.

Response: We sincerely appreciate the reviewer for bringing up this point. In comparison to our prior publication in KDD 2022, there exist key distinctions in the following five aspects within this paper. First, we introduced several fine-tuning strategies in this paper to better leverage the knowledge of the pre-trained model. Furthermore, we extended our benchmarking tests to encompass fifty-two molecular property datasets from the Therapeutics Data Commons (TDC)^{1,2} and MoleculeACE³ benchmarking platforms. This expansion further demonstrated the superior ability of KPGT in predicting molecular properties from various domains. Moreover, comprehensive analyses of the knowledge acquired by KPGT in both the pre-training and finetuning processes were conducted. These analyses revealed the capability of KPGT to capture both structural and semantic information of molecules, while also highlighting its proficiency in delivering meaningful interpretability for its predictions. Additionally, we applied KPGT to identify potential inhibitors for HPK1 and FGFR1, demonstrating the applicability of KPGT in real drug discovery scenarios. We also conducted comprehensive ablation studies to validate the effectiveness of the specific design choices of our proposed method. These enhancements distinguish our current work from our prior KDD 2022 publication.

3. The masked-based self-supervised molecular graph pre-training has been around for a while. The line graph introduced in this paper has already demonstrated its effectiveness at KDD 2022. Thus, the community's contributions to this paper are limited.

Response: As previously addressed in our response to Point 2, there are several significant distinctions between this paper and our prior publication. First, our work conducted comprehensive benchmarking tests on over sixty datasets, as presented in the section titled "KPGT outperforms baseline methods in molecular property prediction." This section showcased the superior performance of our proposed method compared to a diverse range of baseline methods in predicting molecular properties across various domains. Moreover, the detailed analyses in the section "Investigating the knowledge acquired by KPGT in pre-training and finetuning" revealed that KPGT can capture both the structural and semantic information of molecules. This finding illuminated the significance of introducing additional knowledge into the pre-training process, offering a promising avenue for advancing the design of pre-training strategies for molecule property prediction.

Furthermore, we demonstrated the practical applicability of KPGT by applying KPGT to identify potential inhibitors against HPK1 and FGFR1. These practical applications underscore the value of our work in advancing the field of AI-aided drug discovery. In summary, we believe that our work has made sufficient contributions to the field, not only by presenting compelling results but also by providing important insights into the development of self-supervised learning strategies for molecule property prediction. In addition, it offers a robust and effective tool for advancing AI-aided drug discovery.

Reviewer 2

1. The authors have carefully addressed most of the provided comments, while it is still expected to validate the generalization of the proposed model to different targets within the case study if possible. I recommend publishing this paper. Thank you very much!

Response: We are delighted by the positive feedback from the reviewer. In response to the reviewer's valuable suggestions, we will extend the application of our proposed method to identify potential drug candidates for various targets in our future research.

Reviewer 3

1. The authors have addressed all of my concerns.

Response: We would like to express our gratitude to the reviewer for the thorough and insightful comments.

References

1. Kexin Huang, Tianfan Fu, Wenhao Gao, Yue Zhao, Yusuf Roohani, Jure Leskovec, Connor W Coley, Cao Xiao, Jimeng Sun, and Marinka Zitnik. Therapeutics data commons: Machine learning datasets and tasks for drug discovery and development. Proceedings of Neural Information Processing Systems, NeurIPS Datasets and Benchmarks, 2021.
2. Kexin Huang, Tianfan Fu, Wenhao Gao, Yue Zhao, Yusuf Roohani, Jure Leskovec, Connor W Coley, Cao Xiao, Jimeng Sun, and Marinka Zitnik. Artificial intelligence foundation for therapeutic science. Nature Chemical Biology, 2022.
3. Derek van Tilborg, Alisa Alenicheva, and Francesca Grisoni. Exposing the limitations of molecular machine learning with activity cliffs. Journal of Chemical Information and Modeling, 62(23):5938–5951, 2022. PMID: 36456532.